# High-throughput characterization of HLA-E-presented CD94/NKG2x ligands reveals peptides which modulate NK cell activation

Brooke D. Huisman[1,2,7], Ning Guan [1,2,7], Timo Rückert [3], Lee Garner[4], Nishant K. Singh[1,5], Andrew J. McMichael [4], Geraldine M. Gillespie [4], Chiara Romagnani [3,6] & Michael E. Birnbaum [1,2,5]

HLA-E is a non-classical class I MHC protein involved in innate and adaptive immune recognition. While recent studies have shown HLA-E can present diverse peptides to NK cells and T cells, the HLA-E repertoire recognized by CD94/NKG2x has remained poorly defined, with only a limited number of peptide ligands identified. Here we screen a yeast-displayed peptide library in the context of HLA-E to identify 500 high-confidence unique peptides that bind both HLA-E and CD94/NKG2A or CD94/NKG2C. Utilizing the sequences identified via yeast display selections, we train prediction algorithms and identify human and cytomegalovirus (CMV) proteome-derived, HLA-E-presented peptides capable of binding and signaling through both CD94/NKG2A and CD94/NKG2C. In addition, we identify peptides which selectively activate NKG2C+ NK cells. Taken together, characterization of the HLA-E-binding peptide repertoire and identification of NK activity-modulating peptides present opportunities for studies of NK cell regulation in health and disease, in addition to vaccine and therapeutic design.

Human leukocyte antigen E (HLA-E) is a non-canonical, class Ib Major Histocompatibility Complex (MHC) expressed on most nucleated cells in the body. HLA-E has traditionally been thought to function by presenting sequences derived from a highly conserved segment from class I MHC (MHC-I) signal peptides for recognition by NK cells[1,2]. Recognition of HLA-E in complex with these peptides serves as a mechanism to assess "missing self": when pathogens downregulate class I MHC expression, HLA-E surface expression is in turn decreased, resulting in NK cell cytotoxic activity due to the loss of an inhibitory signal[2,3].

HLA-E's cognate receptor heterodimers, the inhibitory CD94/NKG2A and the activating CD94/NKG2C (collectively referred to here as CD94/NKG2x), are expressed by NK cells and certain T cells. CD94/NKG2A is an ITIM motif-containing inhibitory receptor, while CD94/NKG2C is an ITAM motif-associating

activating receptor[4,5]. CD94/NKG2x bind HLA-E in a peptide-dependent manner[6], and NK cells integrate signaling from these and other receptors to modulate their cytotoxic activity[7]. The fraction of NK cells expressing NKG2A and NKG2C varies widely across patients and disease states, with a large dependency on CMV serostatus[8,9]. Moreover, NKG2C+ NK cells have been associated with a lower viral setpoint in HIV infection[9–11] and with a reduced relapse risk in leukemia patients who have undergone hematopoietic stem cell transplantation[12], pointing towards strong anti-viral and anti-tumor activity of NKG2C+ NK cells. CD94/NKG2C and CD94/NKG2A are also expressed by significant fractions of tumor-infiltrating CD8+ T cells[13,14]. Accordingly, modulation of NK and T cell function through CD94/NKG2x receptors is of therapeutic interest; NKG2A has been proposed as

[1]Koch Institute for Integrative Cancer Research, Cambridge, MA, USA. [2]Department of Biological Engineering, MIT, Cambridge, MA, USA. [3]Innate Immunity, Deutsches Rheuma-Forschungszentrum Berlin (DRFZ), ein Leibniz Institut, Berlin, Germany. [4]Centre for Immuno-Oncology, Old Road Campus Research Building, Nuffield Department of Medicine, University of Oxford, Oxford, UK. [5]Ragon Institute of MGH, MIT, and Harvard, Cambridge, MA, USA. [6]Charité - Universitätsmedizin Berlin, Berlin, Germany. [7]These authors contributed equally: Brooke D. Huisman, Ning Guan. ✉e-mail: mbirnb@mit.edu

a possible therapeutically relevant immune checkpoint, and anti-NKG2A monoclonal antibodies alone or in combination promote anti-tumor immunity in mouse models and Phase II clinical trials[15–17].

Beyond the canonical MHC-I-derived peptide ligands, which share a highly conserved motif and are collectively referred to as VL9 peptides, recent work suggests the HLA-E peptide repertoire is more diverse than previously thought, and has motivated study of TCR-mediated responses to HLA-E homologs in other species (MHC-E)[18–20]. Vaccine strategies against SIV, HBV, and typhoid, including CMV-vector-based vaccines, elicit MHC-E-restricted CD8$^+$ T cell responses[19,21–23]. Inducing HLA-E-restricted T cell responses may be a particularly attractive therapeutic strategy due to the essentially invariant nature of HLA-E as compared to class Ia MHCs, allowing for a given peptide therapy to be potentially beneficial for all individuals. Despite interest in HLA-E and CD94/NKG2x for infectious disease and cancer treatment, a limited number of HLA-E peptide binders are currently known, with approximately 700 total contained in the Immune Epitope Database (IEDB)[24], but likely do not represent a systematic assessment of all possible HLA-E binders.

Here, we use a yeast display-based approach to conduct high-throughput analysis of HLA-E-presented CD94/NKG2x ligands. Starting with a randomized library of 100 million unique peptides linked to HLA-E, we select for binding to HLA-E and either CD94/NKG2A or CD94/NKG2C. Deep sequencing of the HLA-E selection libraries identifies ~500 high-confidence unique peptide binders. With these data, we develop computational algorithms to predict proteome-derived peptide binders to HLA-E and NK receptors. To validate a set of potential peptide binders, we perform biophysical validation, including peptide stability assays and surface plasmon resonance (SPR), combined with NK cell activation analysis. We identify human- and CMV-derived peptides able to affect NK cell effector functions, including peptides that selectively lead to NK cell activation. This improved understanding of the HLA-E-presented CD94/NKG2x peptide repertoire could serve as a tool for therapeutic design, and has demonstrated utility for identification of peptides capable of modulating NK cell activity.

## Results
### Design of HLA-E for yeast display and library selections using CD94/NKG2x

To study the diversity of the HLA-E peptide repertoire recognized by CD94/NKG2x, we adapted an MHC yeast surface display system previously used to study class Ia MHC-restricted TCR recognition[25,26]. In this system, peptide, Beta-2-microglobulin (β2M), and MHC heavy chain are covalently linked to each other and fused to the N-terminus of the yeast mating factor protein Aga2p (Fig. 1A, B). The HLA-E single chain trimer construct expressed well on the yeast surface, as confirmed via staining for an included hemagglutinin (HA) epitope tag and via an anti-HLA-E antibody (Fig. 1C). Since epitope tag staining assesses construct expression rather than protein fold[27–29], we tested peptide-HLA-E (pHLA-E) fold by staining with its cognate NK receptors. Using a canonical VL9 signal peptide-derived sequence (VMAPRTLFL), pHLA-E incubated with CD94/NKG2A tetramer showed positively stained cells, indicating pHLA-E was correctly folded for receptor recognition (Fig. 1C). CD94/NKG2C tetramer and dextramer staining of VMAPRTLFL/HLA-E-expressing yeast showed minimal signal, consistent with its lower affinity[30,31] (Supplemental Fig. 1A, E–G), though both receptors were able to bind and enable selective enrichment of VMAPRTLFL/HLA-E-expressing yeast using avid streptavidin-coated beads (Supplemental Fig. 1B–D).

Next, we created a nonameric (9mer) peptide library using the pHLA-E construct. To screen for peptide preferences for both HLA-E and CD94/NKG2x binding, we fully randomized each position along the peptide rather than limiting sequence diversity at the P2/P9 anchor positions as described for previous pMHC libraries[25]. We then iteratively enriched the randomized peptide library displayed by HLA-E for binding to CD94/NKG2A or CD94/NKG2C via magnetic selection (Supplemental Fig. 2A). After three rounds of selecting with biotinylated receptors bound to streptavidin-coated beads, we conducted both iterative and cross-selections using CD94/NKG2x tetramers, which were more stringent due to tetramers' lower avidity (Supplemental Fig. 2B). Cross-selection was performed to study ability of NKG2C-enriched peptides to bind NKG2A, and vice versa. CD94/NKG2A showed similar tetramer staining for yeast previously selected by CD94/NKG2A or CD94/NKG2C for three rounds, indicating that the peptide binding motifs of CD94/NKG2A and CD94/NKG2C may have significant similarities (Supplemental Fig. 2B), consistent with sequence conservation between NKG2A and NKG2C at HLA-E- and peptide-contacting residues[30]. CD94/NKG2C tetramer staining was low in the fourth round of selections, consistent with staining of HLA-E-presented VMAPRTLFL peptide (Supplemental Fig. 1A), though tetramer and dextramer staining, as well as bead-based binding to epitope-tagged post-selection libraries, supports that the libraries have been enriched for binders (Supplemental Fig. 1B–G).

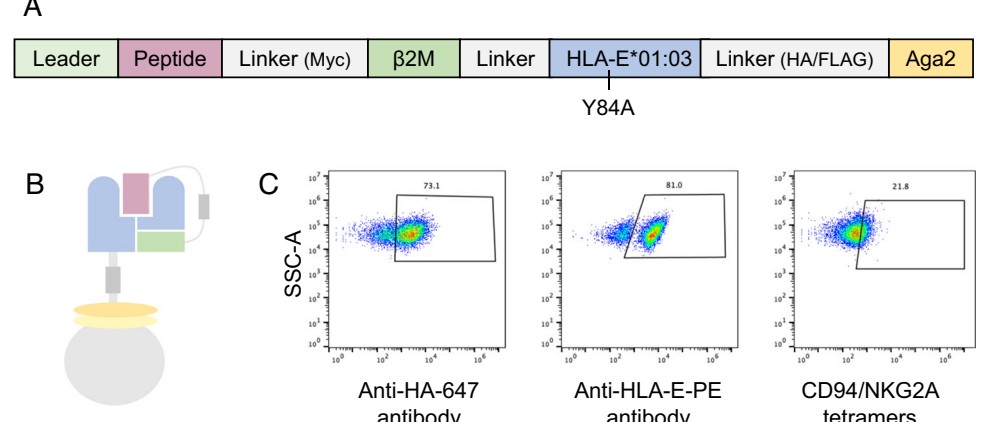

**Fig. 1 | Design and validation of HLA-E on yeast.** Representation of HLA-E (**A**) sequence and (**B**) on the surface of yeast. **C** Validation staining of HLA-E with VMAPRTLFL peptide by anti-HA-647 (against an HA epitope tag in the construct) and anti-HLA-E-PE antibodies and with CD94/NKG2A tetramers made with Streptavidin-647.

## Analysis of library-enriched peptides reveals highly diverse peptides with much overlap between HLA-E-presented CD94/NKG2A and CD94/NKG2C ligands

Following selections, we performed deep sequencing of each round of library-enriched yeast to determine the identities of the enriched peptides (Supplemental Data 1). We identify HLA-E 9mer library hits with significantly greater diversity than the canonical MHC-I leader-like ligands[18,20]. Specifically, deep sequencing showed ~10⁴ unique peptides per sample after the third round, and ~5×10³ unique peptides per sample after the fourth round of selection, although approximately 500 peptides are dominant among the enriched peptides, each with read counts >100 (Supplemental Fig. 3).

A clear motif of enriched peptides emerged in the data after the third round of selection that deviated from the unselected library in a statistically significant manner (Fig. 2 and Supplemental Figs. 4 and 5). The motif, highly conserved between CD94/NKG2A-selected and CD94/NKG2C-selected libraries, contained significant diversity on the peptide N-terminus. Among primary anchors[32], P2 accommodated multiple hydrophobic residues, and P9 strongly preferred Leu. P3 enriched strongly for Pro, with a stronger preference than the traditional P2 anchor residue preferences. As secondary anchors, P6 enriched for Ser and Thr, and P7 enriched for Leu in addition to other hydrophobic residues. We generally observed greater conservation of the peptide C-terminus (P5 through P9) in enriched sequences, likely corresponding to constraints imposed by the CD94/NKG2x-pHLA-E binding interface[30,33]. P5 and P8 are primary CD94/NKG2x-contact residues[34]. Our selection data identified P5 with strong preference for Arg, while P8 displayed preference for Trp, Phe, or Leu. CD94/NKG2A

demonstrated more relaxed binding constraints as compared to CD94/NKG2C, potentially due to CD94/NKG2A's higher affinity for HLA-E[30]. While the motifs after Round 4 selection with CD94/NKG2x tetramers were consistent with Round 3 motifs, Round 4 CD94/NKG2A tetramer selection led to a less diverse motif likely containing higher affinity binders (Supplemental Fig. 4, Post-Round 4: CA and Post-Round 4: A). Motifs of peptides after selections with CD94/NKG2C tetramer in Round 4 were similar to post-Round 3 motifs, suggesting the low avidity of the CD94/NKG2C tetramer did not further enrich the library, consistent with prior reports on tetramer binding of low-affinity receptors[31]. Through peptide clustering, we also observed a subset of peptides characterized by strong preference for P1 Trp and P2 Asn ("WN peptides") with amino acid preferences at the N-terminus of the peptide yet little constraint on the C-terminal portion of the peptide (Supplemental Fig. 6).

While the majority of described HLA-E-restricted peptides have thus far been 9 amino acids long[24], class Ia MHCs can bind a range of additional peptide lengths, with 10mers being nearly as prevalent as 9mers for some alleles[35]. Therefore, in addition to the 9mer library, we also generated a 10mer peptide library linked to HLA-E and performed selections with CD94/NKG2C. Interestingly, the motif for the first 9 amino acids of the 10mer largely resembled the motif in the 9mer library (Supplemental Fig. 7). The C-terminal amino acid showed less preference than the 9mer library PΩ position, consistent with the 10mer peptides binding as 9mer peptides, with the tenth residue acting as part of the peptide linker for the single chain trimer-formatted pMHC, with subtle preferences potentially resulting from interactions outside of the peptide binding groove. This suggests that 9mer

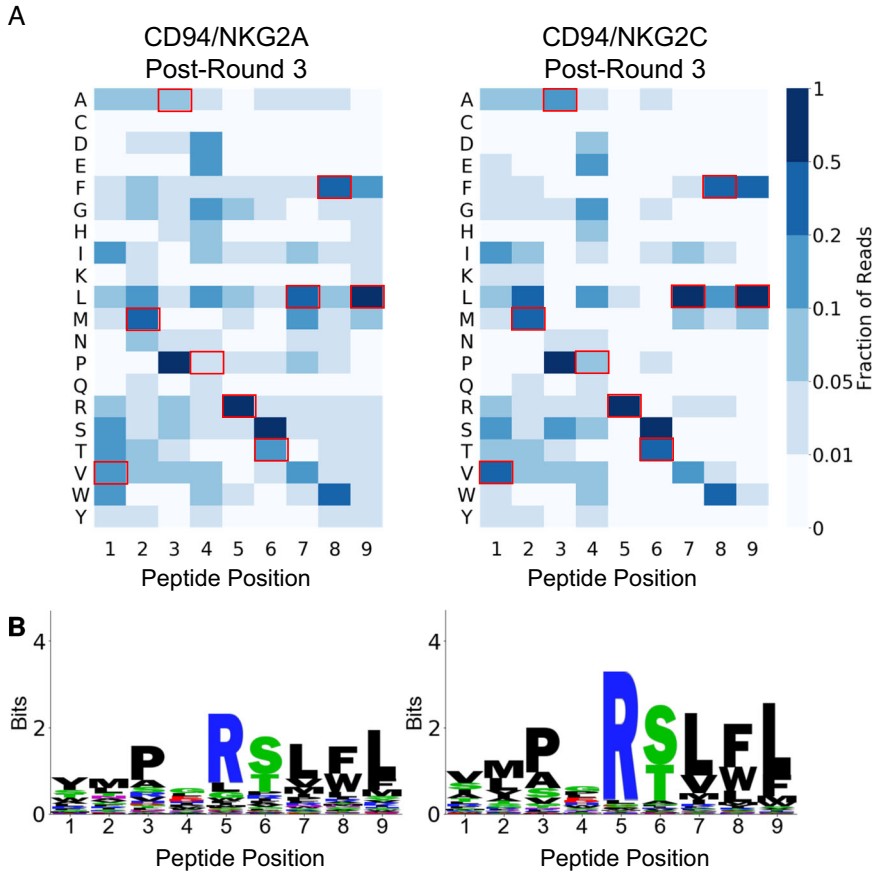

**Fig. 2 | Repertoire of HLA-E-presented peptides that bind to CD94/NKG2A or CD94/NKG2C. A** Heatmaps showing peptide positional amino acid preferences for binding HLA-E and CD94/NKG2A or CD94/NKG2C with peptides weighted by read count. Residues corresponding to the VMAPRTLFL VL9 peptide are highlighted with red boxes. **B** Sequence logos present the same data, also weighted by read count. Source data are provided as a Source Data file, and peptide sequences are provided in Supplemental Data 1.

**Table 1 | VL9 peptide predicted ranks**

| Peptides | NKG2A model: rank | NKG2C model: rank |
|---|---|---|
| VMAPRTLFL | 224 | 2 |
| VMAPRTLLL | 418 | 12 |
| VMAPRTVLL | 547 | 22 |
| VMAPRTLIL | 481 | 32 |
| VMAPRTLVL | 602 | 48 |
| VMAPRALLL | 514 | 69 |

Predicted ranks of eight signal peptides by CD94/NKG2A and CD94/NKG2C models, ranked out of 11 million human-derived 9mer peptides.

peptides preferentially bind to CD94/NKG2x in the context of HLA-E compared to 10mer peptides, leading us to focus on the 9mer library data.

### Training prediction algorithms and predicting human- and pathogen-derived ligands

Given the large theoretical space ($20^9$) of 9mer peptides, yeast-displayed peptide libraries do not comprehensively examine all possible peptide sequences. We therefore trained a prediction algorithm using our yeast display-derived peptide data to predict putative human- and pathogen-derived peptide binders to HLA-E and CD94/NKG2x. We used yeast selection data to train the NNAlign architecture[36], which we have previously shown produces high-confidence class II MHC-peptide binding predictions using yeast display-derived data[29]. We trained separate models using yeast display data from Round 3 of CD94/NKG2A and CD94/NKG2C selections, using a stringent read cutoff, resulting in 376 and 399 unique positive examples for NKG2A- and NKG2C-specific predictors, respectively, with negative examples drawn from the unselected library.

We applied this algorithm to the 11 million unique 9mers, generated from a sliding window of size 9, step size 1, along a reference human proteome. Given the absence of relevant comparator methods for predicting HLA-E and CD94/NKG2x binding, we examined the rankings of known binders. The ranks of the following canonical signal peptide-derived HLA-E ligands were examined: VMAPRTLFL, VMAPRTLLL, VMAPRTLIL, VMAPRALLL, VMAPRTVLL, VMAPRTLVL[37–44], with their predicted ranks shown in Table 1. The distribution of ranks for these peptides were significantly better than the human proteome as a whole (with $p$-value of $1.1 \times 10^{-5}$ by Mann–Whitney U test, for either CD94/NKG2A and CD94/NKG2C model predictions).

The top ten predicted human proteome-derived peptides are shown in Table 2. These binders largely share features of known binders, including hydrophobic P7-P9 residues and P5 Arg. One of the top predicted binders resembles the N-terminal motif of the subdominant WN peptides (UBAC2$_{275-279}$).

In addition to predicting human-derived peptides, we applied our models to a reference human CMV proteome. One mechanism by which CMV evades T cell and NK cell detection is by generating a UL40-derived peptide identical to canonical VL9 signal peptides, which can be loaded onto HLA-E, even when canonical class I processing is disrupted[37,44–46]. We ranked a UL40-derived peptide highly among CMV-derived peptides (Table 3). Interestingly, however, both CD94/NKG2A and CD94/NKG2C models rank a UL120-derived peptide VLPHRTQFL (UL120$_{72-80, \text{Merlin}}$) more highly, which bears the hallmarks of a strong binder, including P3 Pro, P5 Arg, P8 Phe, and P9 Leu. Our predicted peptide UL120$_{72-80, \text{Merlin}}$ is from a highly polymorphic region among different CMV strains. For example, despite a large degree of conservation in the UL120 proteins between strain Merlin (used in our search) and related strains, strain AD169 has the divergent peptide SAPLKTRFL ("UL120$_{71-79, \text{AD169}}$"), and strain BE/33/2010 has the divergent peptide SVPLKTRFL ("UL120$_{71-79, \text{BE/33/2010}}$").

When comparing our model predictions on these peptides to the rankings of 61,303 CMV-derived peptides, UL120$_{71-79, \text{AD169}}$ ranks after the 104th (NKG2A model) and 14th peptides (NKG2C model), and UL120$_{71-79, \text{BE/33/2010}}$ ranks after the 20th (NKG2A model) and 25th peptides (NKG2C model), although predicted binding scores drop sharply within the top ten peptides.

### Validating peptides for HLA-E binding ability

To probe top-predicted proteome-derived peptides (Tables 2 and 3) for HLA-E binding, we assessed their ability to stabilize surface-expressed HLA-E on RMA-S/HLA-E cells, a TAP-deficient mouse cell line engineered to express HLA-E and human β2M (Fig. 3 and Supplemental Fig. 8)[47]. Ten human peptides showed strong stabilization of HLA-E (Human peptides INTS1$_{260-268}$, HLA-A$_{3-11}$, ECEL1$_{269-277}$, TACR3$_{226-234}$, CREB3L1$_{419-427}$, AKAP6$_{388-396}$, MTREX$_{490-498}$, FBXO41$_{670-678}$, SLC52A3$_{354-362}$, PISD$_{55-63}$). An additional two human peptides demonstrated weak but detectable stabilization (Human peptides BFAR$_{263-271}$, GTF3C5$_{293-301}$). Of the CMV peptides, UL120$_{72-80, \text{Merlin}}$ showed strong HLA-E binding, and peptide variants in different viral strains (UL120$_{71-79, \text{AD169}}$, UL120$_{71-79, \text{BE/33/2010}}$) showed weaker but detectable stabilization. Results are consistent with thermal stability measurements taken on a subset of human and CMV proteome-derived predicted peptides (Supplemental Fig. 9), with relative melting temperatures between peptides matching relative cell surface stabilization effects. Several peptides showed no stabilizing effects (Human peptides UBAC2$_{275-279}$, EMC1$_{725-733}$, OR5D14$_{88-96}$, and CMV peptide UL102$_{292-300}$) (Supplemental Fig. 10). EMC1$_{725-733}$, OR5D14$_{88-96}$, and CMV peptide UL102$_{292-300}$ contain P7 Phe and P9 Tyr, which are absent from stabilizing peptides, and also lack P3 Pro, suggesting these residues or combinatorial effects of these residues may have been improperly weighted by the prediction algorithm. UBAC2$_{275-279}$ contains an N-terminal WN motif, and we tested additional peptide variants containing this previously identified motif (Supplemental Fig. 6) but observed no detectable stabilization (Supplemental Fig. 10), suggesting enrichment of WN peptides is a result of the peptides being covalently linked in the single chain trimer system.

### Proteome-derived peptides demonstrate ability to modulate NK cell activation

We hypothesized that our predicted proteome-derived peptides could affect NK cell activation through interactions with CD94/NKG2A and CD94/NKG2C. To investigate, we performed cell-based activation assays on primary NK cells (Fig. 4A). NK cells were co-incubated with peptide-pulsed HLA-E-expressing K562 cells, as described previously[47]. The parental K562 human myelogenous leukemia cell line is a typical target to measure NK cell cytotoxicity and activation[47], allowing for assessment of both NKG2A-mediated inhibition and NKG2C-mediated activation. A side-by-side comparison of the HLA-E expression levels of K562/HLA-E and RMA-S/HLA-E cell lines is shown in Supplemental Fig. 11. All peptides which bound and stabilized HLA-E in the cell-based stability experiment (Fig. 3) were analyzed for their effects on NKG2A$^+$/NKG2C$^-$ or NKG2A$^-$/NKG2C$^+$ NK cell activity, as measured by their ability to induce NK cell degranulation as well as IFN-γ, TNF and CCL3 production, in comparison to class I MHC-derived positive control peptides VMAPRTLIL and VMAPRTLFL, and negative control peptide VMAPQSLLL (Fig. 4; with coloring as in Fig. 3)[47]. NKG2A$^-$/NKG2C$^-$ NK cells served as internal control as these should be unaffected by peptide-HLA-E complexes (Supplemental Fig. 12). Six peptides had both strong inhibitory effects on NKG2A$^+$ cells and strong activating effects on NKG2C$^+$ cells (Human peptides INTS1$_{260-268}$, HLA-A$_{3-11}$, ECEL1$_{269-277}$, TACR3$_{226-234}$, MTREX$_{490-498}$; CMV peptide UL120$_{72-80, \text{Merlin}}$), and four peptides had more moderate inhibitory effects on NKG2A$^+$ cells while maintaining strong activating effects on NKG2C$^+$ cells (Human peptides CREB3L1$_{419-427}$, AKAP6$_{388-396}$, FBXO41$_{670-678}$, SLC52A3$_{354-362}$).

**Table 2 | Top ten predicted human peptides from both models**

| Model | Rank | Peptide | Source | Name |
|---|---|---|---|---|
| NKG2A | 1 | QMPSRSLLF | Cyclic AMP-responsive element-binding protein 3-like protein 1 | CREB3L1$_{419-427}$ |
| NKG2A | 2 | TLPKRGLFL | A-kinase anchor protein 6 | AKAP6$_{388-396}$ |
| NKG2A | 3 | TGPWRSLWI | General transcription factor 3C polypeptide 5 | GTF3C5$_{293-301}$ |
| NKG2A | 4 | ILTDRSLWL | F-box only protein 41 | FBXO41$_{670-678}$ |
| NKG2A | 5 | VNPGRSLFL | Bifunctional apoptosis regulator | BFAR$_{263-271}$ |
| NKG2A | 6 | WNRLFPPLR | Ubiquitin-associated domain-containing protein 2 | UBAC2$_{275-279}$[a] |
| NKG2A | 7 | TLPERTLYL | Endothelin-converting enzyme-like 1 | ECEL1$_{269-277}$ |
| NKG2A | 8 | FLPNRSLLF | Solute carrier family 52, riboflavin transporter, member 3 | SLC52A3$_{354-362}$ |
| NKG2A | 9 | VMGDRSVLY | ER membrane protein complex subunit 1 | EMC1$_{725-733}$ |
| NKG2A | 10 | VMADKSIFY | Olfactory receptor 5D14 | OR5D14$_{88-96}$ |
| NKG2C | 1 | VMPPRTLLL | HLA class I histocompatibility antigen | HLA-A$_{3-11}$ |
| NKG2C | 2 | VMAPRTLFL | HLA class I histocompatibility antigen | VMAPRTLFL (control) |
| NKG2C | 3 | VNPGRSLFL | Bifunctional apoptosis regulator | BFAR$_{263-271}$ |
| NKG2C | 4 | QMPSRSLLF | Cyclic AMP-responsive element-binding protein 3-like protein 1 | CREB3L1$_{419-427}$ |
| NKG2C | 5 | TLPERTLYL | Endothelin-converting enzyme-like 1 | ECEL1$_{269-277}$ |
| NKG2C | 6 | VMPGRTLCF | Neuromedin-K receptor | TACR3$_{226-234}$ |
| NKG2C | 7 | ILTDRSLWL | F-box only protein 41 | FBXO41$_{670-678}$ |
| NKG2C | 8 | RMPPRSVLL | Integrator complex subunit 1 | INTS1$_{260-268}$ |
| NKG2C | 9 | TAPARTMFL | Phosphatidylserine decarboxylase proenzyme, mitochondrial | PISD$_{55-63}$ |
| NKG2C | 10 | NMPARTVLF | Exosome RNA helicase MTR4 | MTREX$_{490-498}$ |

Top predicted peptides by CD94/NKG2A and CD94/NKG2C models, with source proteins noted.
[a]UBAC2$_{275-279}$ peptide was assessed for cell-based testing as a 5mer peptide (WNRLF).

**Table 3 | Top predicted CMV peptides from both models**

| NKG2A model: rank | NKG2C model: rank | Peptide | Source | Name |
|---|---|---|---|---|
| 1 | 1 | VLPHRTQFL | Membrane protein UL120 | UL120$_{72-80, \text{Merlin}}$ |
| 2 | 3 | TGAARSFFF | DNA helicase/primase complex-associated protein | UL102$_{292-300}$ |
| 3 | 2 | VMAPRTLIL | UL40 | VMAPRTLIL (control) |

Top predicted peptides by CD94/NKG2A and CD94/NKG2C models, with source protein noted. Ranks are among 61 thousand CMV-derived peptides.

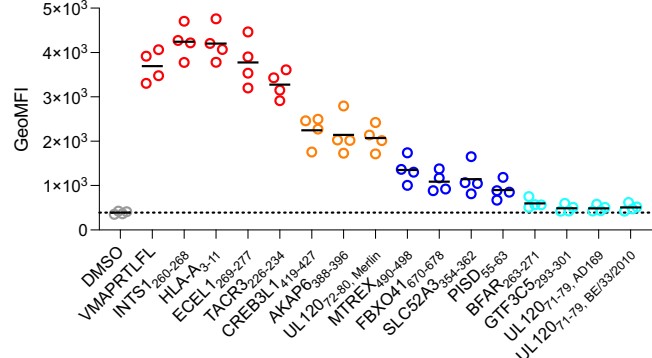

**Fig. 3 | Peptide stabilization of HLA-E surface expression.** Assessment of peptide-HLA-E binding via HLA-E surface stabilization assay with RMA-S/HLA-E cells incubated with 30 µM peptide. HLA-E expression is detected with an anti-HLA-E antibody. Measurements from n = 4 replicate experiments are shown, with solid black lines indicating mean values. Peptides with similar stabilization effects are plotted in the same color; in order from most to least stabilizing, the colors are: red, orange, blue, and cyan, with DMSO control in gray. Source data are provided as a Source Data file.

nonspecific effects. UL120$_{71-79, \text{BE/33/2010}}$ peptide showed minimal effects on both NKG2A$^+$ or NKG2C$^+$ cells.

In contrast to K562 cell lines which are targets for human NK cells, RMA-S cells (a TAP-deficient mouse cell line)[48] are inert to human NK cells and thus will not activate NK cells absent engagement of an activating receptor on the NK cells. Because of this, RMA-S/HLA-E cells have been shown to provide additional granularity on the strength of NKG2C-mediated NK cell activation[47]. We performed additional NK cell activation assays with RMA-S/HLA-E cells (Fig. 4B). Minimal activation is observed for the negative control peptide (VMAPQSLLL) and differential activation is observed for positive control VMAPRTLIL and VMAPRTLFL peptides, consistent with previous data[47]. In previous studies, VMAPRTLIL and VMAPRTLFL peptides have demonstrated differential activating effects, with VMAPRTLFL peptide acting as the most potent activating peptide, which we observe as well (Fig. 4B)[47]. Remarkably, two peptides (Human peptides AKAP6$_{388-396}$, FBXO41$_{670-678}$) showed activation at the level of the strongest positive control binder (VMAPRTLFL) (Fig. 4B). Additional peptides showed activation similar to the weaker positive control (VMAPRTLIL) or between the two positive controls (Human peptides HLA-A$_{3-11}$, ECEL1$_{269-277}$, TACR3$_{226-234}$, CREB3L1$_{419-427}$, SLC52A3$_{354-362}$, BFAR$_{263-271}$; CMV peptide UL120$_{72-80, \text{Merlin}}$). Three peptides showed detectable activation, though below both positive control examples (Human peptides MTREX$_{490-498}$, PISD$_{55-63}$, GTF3C5$_{293-301}$). In sum, among our predicted peptides, we have identified sequences able to modulate NK cell activity at the level of positive control ligands, and have additionally identified peptides

Strikingly, three peptides showed minimal inhibitory effects but clear activating effects (Human peptides PISD$_{55-63}$, BFAR$_{263-271}$, GTF3C5$_{293-301}$). The UL120$_{71-79, \text{AD169}}$ peptide seemed to negatively affect activation of all subsets, including the internal NGK2A$^-$ NKG2C$^-$ control, suggesting

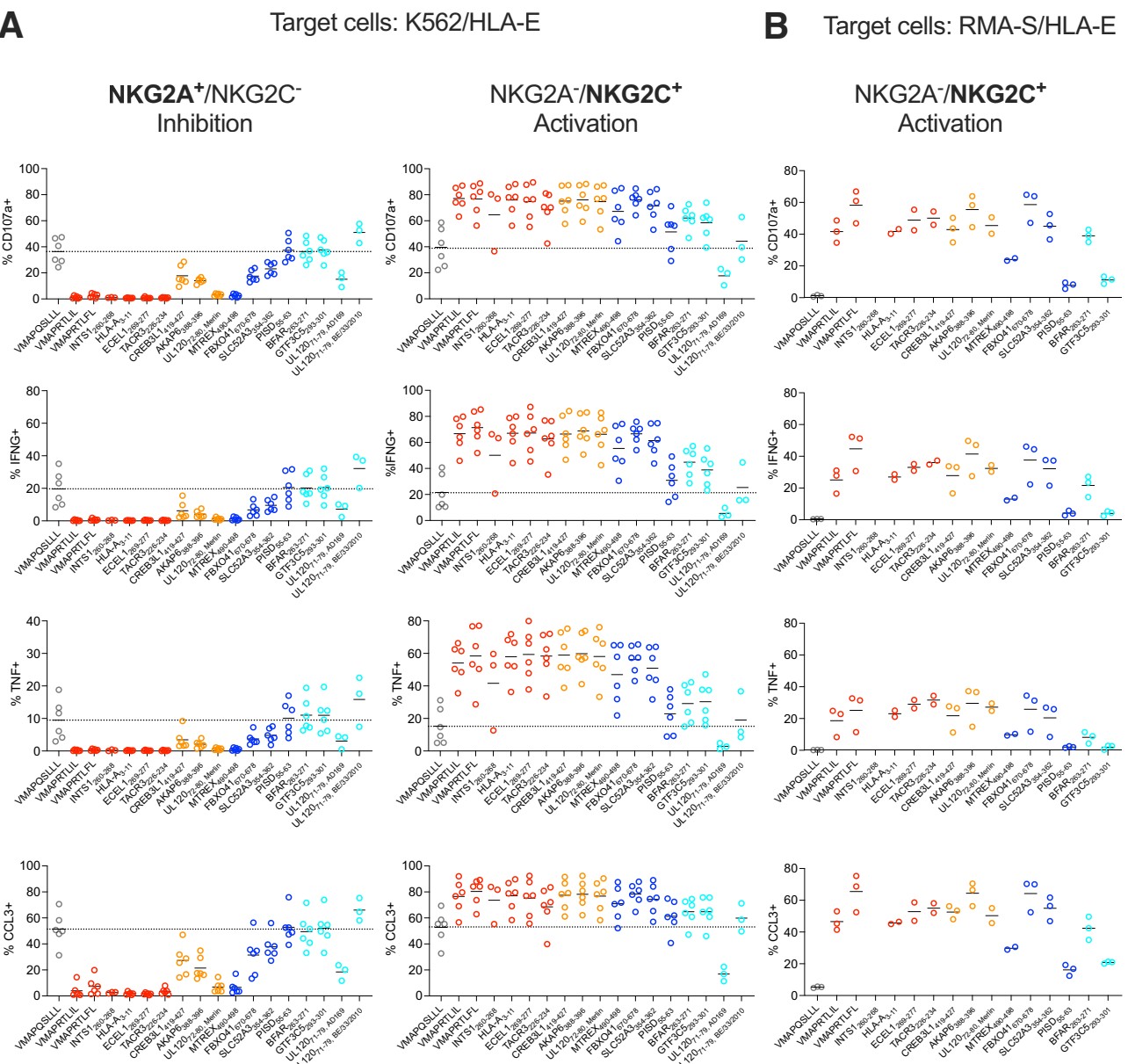

**Fig. 4 | NK cell stimulation assays.** Assessment of effects of peptides on NK cell activity through incubation of NK cells, peptides, and (**A**) K562/HLA-E or (**B**) RMA-S/HLA-E target cells. Peptides were derived from the human or CMV proteomes or are included as positive (VMAPRTLFL, VMAPRTLIL) or negative controls (VMAPQSLLL). Replicates for individual peptides are from different donors, with solid black lines indicating mean values. Stabilization experiments with K562/HLA-E cells were performed with $n = 6$ biologically independent samples excepting $n = 3$ for INTS1$_{260-268}$, UL120$_{71-79, AD169}$, and UL120$_{71-79, BE/33/2010}$, and stabilization experiments with RMA-S/HLA-E cells were performed with $n = 3$ biologically independent samples excepting $n = 2$ for ECEL1$_{269-277}$, TACR3$_{226-234}$, HLA-A$_{3-11}$, and UL120$_{72-80, Merlin}$. Peptides are colored as in Fig. 3, highlighting stabilization effects; in order from most to least stabilizing, the colors are: red, orange, blue, and cyan, with the negative control in gray. Representative gating for NKG2A$^+$/NKG2C$^-$ and NKG2A$^-$/NKG2C$^+$ NK cells is shown in Supplemental Fig. 14. Source data are provided as a Source Data file.

that are able to selectively activate NK cells, with minimal inhibitory effects.

## Measuring pHLA-E-CD94/NKG2x affinity for identified peptides
To investigate the mechanism by which our activity-modulating peptides selectively activate via CD94/NKG2C without inhibiting through CD94/NKG2A, we measured the affinity of peptide-HLA-E for NK receptors via SPR (Fig. 5). Specifically, we measured CD94/NKG2A and CD94/NKG2C affinity for HLA-E complexed with three sets of peptides. The first set are the most selectively activation-enhancing peptides (Human peptides GTF3C5$_{293-301}$, BFAR$_{263-271}$, and PISD$_{55-63}$) which exhibited NKG2C-mediated activation and minimal NKG2A-mediated

inhibition (Fig. 4). The second set of peptides (Human peptides CREB3L1$_{419-427}$ and AKAP6$_{388-396}$) exhibited both activation and inhibition, through more modest inhibition compared to positive control peptides (Fig. 4), as well as greater HLA-E stability than Human peptides GTF3C5$_{293-301}$, BFAR$_{263-271}$, and PISD$_{55-63}$. Third, we included VMAPRTLFL peptide as a positive control.

All six peptides assessed exhibited higher affinity for CD94/NKG2A than CD94/NKG2C, suggesting that selective activation was not due to higher affinity for CD94/NKG2C (Fig. 5A, C). Further, the selectively activating peptides BFAR$_{263-271}$ and GTF3C5$_{293-301}$ were the highest and lowest affinity peptides for both receptors, respectively, suggesting that selective activation was also not due to an affinity

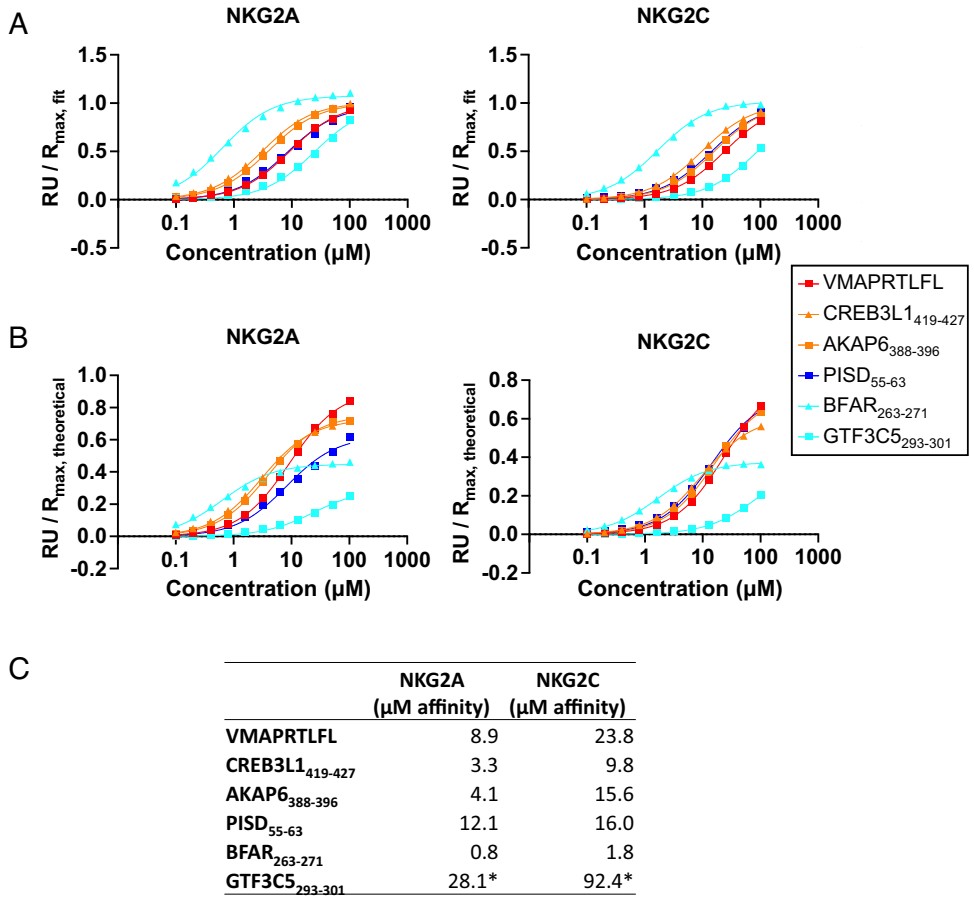

**Fig. 5 | SPR of proteome-derived peptides with CD94/NKG2A and CD94/NKG2C.** Affinity measurements of peptide-HLA-E for CD94/NKG2A and CD94/NKG2C, (**A**) normalized to fitted $R_{max}$ or (**B**) theoretical $R_{max}$. Peptides are colored as in Fig. 3, highlighting stabilization effects. **C** Summary of approximate $K_D$ affinity values for CD94/NKG2x binding to peptide-HLA-E. *Indicates weak binders not approaching saturation, so $K_D$ values are inexact. Source data are provided as a Source Data file.

threshold for signaling that differed between the two receptors. However, when comparing maximal response units ($R_{max}$) to the theoretical maximum that would be reached if all immobilized protein species were bound (Fig. 5B), we observe GTF3C5$_{293-301}$ and BFAR$_{263-271}$ reach lower maximal values compared to the theoretical maximum, including at saturation (BFAR$_{263-271}$). This suggests that GTF3C5$_{293-301}$ and BFAR$_{263-271}$ in complex with HLA-E are less stable, consistent with their HLA-E stabilization ability (Fig. 3).

We additionally assessed peptides containing the N-terminal WN motif for their ability to bind CD94/NKG2x in the context of HLA-E. Though WN motif-containing peptides showed minimal ability to stabilize cell-expressed HLA-E (Supplemental Fig. 10), a WN peptide expressed in a single chain trimer format with HLA-E was able to bind to CD94/NKG2A and CD94/NKG2C (Supplemental Fig. 6), suggesting that the observed enrichment in the yeast display results was a bona fide interaction, but that the peptide-MHC interaction was sufficiently weak as to require covalent linkage to detect CD94/NKG2x binding.

## Discussion

HLA-E has been shown to present diverse peptides to NK and T cell receptors[19,21,23]. This potential for peptide diversity, coupled with its sequence conservation across global populations[49], has led to therapeutic interest in HLA-E and its cognate receptors[13,15,17,21,23]. Despite this interest, the HLA-E-presented NK receptor-recognized peptide repertoire has remained poorly characterized. Here, we utilized yeast display libraries to characterize the repertoire of ligands that can bind to CD94/NKG2x in the context of HLA-E. Utilizing these data, we trained prediction algorithms and identified human- and CMV-derived

peptides which can alter NK cell activity through CD94/NKG2x when presented by HLA-E. While many peptides can both activate through CD94/NKG2C and inhibit through CD94/NKG2A, a subset of peptides selectively activate CD94/NKG2C⁺ NK cells. Interestingly, these selectively activating peptides exhibited less HLA-E stabilization while showing comparable affinity to CD94/NKG2x, suggesting that altering peptide affinity to HLA-E, rather than the pHLA-E affinity to CD94/NKG2x, may be a mechanism for modulating NK cell activity. These effects may be due to potential differences in the level of receptor occupancy required to induce CD94/NKG2A vs CD94/NKG2C signaling, which may be linked to differences in the receptors' signaling apparatuses[5].

Past characterization of the HLA-E peptide repertoire has varied widely, including primary anchor residue locations and preferences[50] and secondary anchor residue preferences[34], likely limited in part by study throughput. Our dataset of HLA-E and CD94/NKG2x-binding peptides is consistent with known high-confidence HLA-E binders and structural knowledge of the interface[30,33], while further characterizing the peptide repertoire. Notably, we identify a strong preference for Pro at position 3, which VL9 peptides contain at position 4 and has been reported to be preferred at positions 3, 4, 6, or 7[32]. Other recent work includes a study from Ruibal et al., utilizing a combinatorial peptide library for screening peptides for HLA-E binding[47] which allows for the interrogation of peptide preferences, although it does not determine the identity of individual peptides. The yeast display approach allows us to determine the identities of individual peptides, which were useful here as training examples for prediction algorithms. Our datasets match at several positions, such as P9, where we both identify Phe and

Leu as the two most preferred residues. However, important differences between the characterizations of HLA-E-binding residues are also present. For example, Ruibal et al. identified a strong preference for P2 Met. In contrast, our dataset suggests a less extreme preference, supported by HLA-E binding of peptides which include P2 Leu, Asn, and Ala, and a more dominant preference for P3 Pro. The NetMHCpan-4.1 motif viewer also highlights a prominent P3 Pro preference for peptides bound to HLA-E*01:01 or HLA-E*01:03, alongside similar amino acid preferences to those captured in our motif at P2 and P9[51], corroborating our motif at these positions.

The peptide preferences that we identify are a composite of preferences for binding to HLA-E and CD94/NKG2x. Preferences at P5 and P8 are consistent with structural studies showing interaction between these residues and CD94/NKG2A[30,33]. Preferences at P2 and P9 are consistent with constraints for HLA-E-binding anchors[30,32,33]. The strong P3 proline preference likely helps establish peptide conformation within the groove and may be helpful for HLA-E binding given its prevalence in the NetMHCpan4.1 motif[51]. Extension of our dataset for identifying potential HLA-E-presented T cell ligands would require excluding NK receptor-specific preferences, and characterization of the HLA-E binding peptide repertoire alone is of interest. We and others have utilized yeast display to characterize the peptide-MHC-II binding repertoire[29,52–55]. However, differences in the closedness of the MHC binding groove and the length of bound peptides have limited their application to MHC-I. Extension of yeast display, or application of complementary platforms such as the recently described EpiScan[56], to study the peptide-HLA-E repertoire may be of subject of future studies.

To format HLA-E for yeast display, we introduced a Y84A mutation, which opens the peptide binding groove to accommodate the peptide linker. This mutation could affect the binding preferences of the MHC F-pocket, but has been shown to largely recapitulate the endogenous binding properties of wild type class I MHCs[25]. This mutation, however, allows the C-terminus of the peptide to extend beyond the peptide binding groove, which appears to be occurring in the 10mer library, suggesting HLA-E preferentially binds 9mer peptides. Assessments of the repertoire of HLA-E-presented peptides which can be recognized by KIRs would be of future interest, although the C-terminal binding of KIRs may be disrupted by the opened peptide groove[57], which may necessitate an alternate peptide linking strategy.

Using our yeast display-generated data, we trained algorithms to predict proteome-derived peptides for binding HLA-E and CD94/NKG2x, using a stringent set of top positive examples and negative examples drawn from the unselected library. While there exist no algorithms for predicting peptide binding to both HLA-E and CD94/NKG2x, NetMHCpan4.1 can be utilized to predict peptide-HLA-E binding. NetMHCpan4.1 predicts that all of our proteome-derived 9mer peptides assessed in Fig. 3 and Supplemental Fig. 10 will bind to HLA-E, either as strong or weak binders[51], including peptides which showed no HLA-E stabilizing effects. As such, there is a high degree of concordance between our predicted binders, which is consistent with us capturing HLA-E-binding preferences; however, NetMHCpan4.1 does not predict CD94/NKG2x binding and cannot prioritize these binders among other predicted peptides.

Our prediction tool enabled us to identify peptides capable of binding to HLA-E and modulating NK cell activity. We used our yeast display-generated data to train the existing NNAlign model architecture[36] which has previously been shown useful for peptide-MHC binding prediction[29,51]. The data generated by yeast display could be used to train future models which may improve predictions via incorporation of features such as improved weighting of residue covariation. For example, three predicted but non-stabilizing peptides (EMC1$_{725-733}$, OR5D14$_{88-96}$, and UL102$_{292-300}$) lack P3 Pro, a residue that is present in all of the stabilizing peptides, except for FBXO41$_{670-678}$, which contains ideal residues at each of the remaining positions.

Improved weighting of positional residue covariation, such as a requirement for P3 Pro in most peptides, may improve the predictive performance of these models.

Newly characterized proteome-derived peptides which affect NK cell activity may present opportunities for studying NK cell regulation in health and disease, and in vaccine and therapeutic design. The identified CMV-derived peptide from the UL120 protein can alter NK cell activity, expanding the known CMV peptides able to bind HLA-E and CD94/NKG2x beyond known UL40-derived peptides[44]. Additionally, because related CMV strains are highly dissimilar at this region, with these peptides demonstrating different functional effects, future work may investigate whether these differences between strains lead to differential HLA-E presentation of CMV-derived peptides and if they contribute to varying responses to CMV strains[45,58]. Similarly, our studies on the human proteome reveal an unappreciated complexity of self-peptides capable of stabilizing HLA-E and modulating NK cell responses, including proteins associated with cancer metastasis and apoptosis regulation[59–61], and future studies may investigate if these peptides provide a more extensive monitor of cellular states through surveillance by NK and potentially T cells of cellular processes beyond MHC class I expression.

Prior work to elucidate peptide processing and presentation on HLA-E using mass spectrometry have been limited by the abundance of VL9 peptides, which may obscure less abundant ligands[62,63]. Although true binders have been characterized among the detected non-VL9 fraction of peptides, this fraction contains high levels of noise, such as long and highly charged peptides[41,62]. Since our study shows the ability for peptides to bind to HLA-E but cannot examine natural peptide presentation, further characterization of the naturally presented HLA-E-peptidome and cross-comparison with the HLA-E-presented CD94/NKG2x ligands identified here will be of interest as mass spectrometry approaches continue to progress.

Characterization of HLA-E-presented CD94/NKG2x ligands enabled the identification of peptides capable of selectively activating NK cells through NKG2C, without inhibiting through the higher-affinity NKG2A. Features of these peptides, including their less preferred P2 residues for HLA-E binding but preferred CD94/NKG2x-facing residues, could be used for future identification or design of additional selectively activating peptides. Recent work has also utilized engineered feeder cells expressing cytokines and VL9/HLA-E to expand NK cells, including selective expansion of CD94/NKG2C[+] NK cells[64]. We present a complementary and versatile peptide-driven method for selective CD94/NKG2C[+] NK cell activation. Given the important role of CD94/NKG2x receptors in controlling NK cell function, and the association of adaptive NKG2C[+] NK cells with viral control[9–11] and anti-leukemic effects[12], these findings may present new potential strategies for modulation of NK cell activity.

Finally, previous studies have identified T cell responses to MHC-E presented peptides without obvious shared features[19]. Through stability assays, we observed that selectively activity-enhancing peptides exhibit less HLA-E stabilizing effects, suggesting NKG2C has a higher tolerance for less stable pHLA-E, and the stability or conformation of HLA-E may play a role in activating T cells or NK cells.

## Methods

All research was carried out in compliance with the relevant ethical regulations.

### Yeast-displayed pMHC design

Yeast-displayed HLA-E*01:03 was generated as a single chain trimer, covalently linked to a peptide and human β2M. Specifically, the C-terminus of the peptide was linked to N-terminus of β2M via Gly-Ser linker containing a Myc epitope tag (EQKLISEEDL) and protease site (LEVLFQGP). A second Gly-Ser linker connected the C-terminus of β2M to the N-terminus of HLA-E heavy chain α1, α2, and α3 domains. HLA-E

C-terminus is connected to Aga2 via a final Gly-Ser linker containing an epitope tag (HA in clonal constructs: YPYDVPDYA; FLAG in library: DYKDDDDK). Yeast display constructs were generated in the pYAL vector, with the Aga2 leader sequence. Yeast strains were grown to confluence at 30 °C in pH 5 SDCAA yeast media then subcultured into pH 5 SGCAA media at $OD_{600} = 1.0$ for 48–72 h induction at 20 °C[65].

### Tetramer, dextramer, and antibody staining yeast
To stain peptide-HLA-E (pHLA-E) with CD94/NKG2x tetramers, each biotinylated receptor was mixed with streptavidin coupled to Alexa-Fluor647 (SAV-647, made in-house) at a 5:1 ratio and incubated for 5 min on ice. Yeast were stained with 500 nM tetramer for 2 h at 4 °C in the dark with rotation. Then, the yeast were washed twice with FACS buffer (0.5% BSA and 2 mM EDTA in 1x PBS) before analysis via flow cytometry (Accuri C6 flow cytometer, BD Biosciences; Franklin Lakes, New Jersey). Dextramer staining was performed as with tetramers, excepting CD94/NKG2x protein was incubated with 25 nM biotinylated dextran 70,000 MW (Thermo Fisher) for five minutes before addition of SAV-647. Tetramer staining coupled an anti-streptavidin antibody was performed as with tetramers, excepting 250 nM PE anti-streptavidin antibody (Biolegend) was added to tetramer solution before adding to yeast.

Antibody staining was performed on yeast washed into FACS buffer, with antibody at a 1:50 volume ratio. Yeast incubated with antibody for at least 20 min and excess antibody was removed by washing with FACS buffer. Staining was assessed on the Accuri C6 flow cytometer. Antibody clone 3D12 was used for anti-HLA-E staining.

### Yeast column enrichment competition assay
A yeast column enrichment competition assay was performed to evaluate pHLA-E binding to CD94/NKG2A receptor. Induced yeast expressing pHLA-E SCT were stained with an anti-epitope tag antibody (HA-AlexaFluor647 for VMAPRTLFL/HLA-E or FLAG-PE for post-selection libraries), then mixed with irrelevant non-HLA-E-expressing yeast without these stains at a 1:5 ratio ($10^6$ HLA-E-expressing yeast, $5 \times 10^6$ irrelevant yeast). A sample of the mixture was assessed via flow cytometry analysis. The remaining mixed population of yeast were incubated with high avidity CD94/NKG2x-coated magnetic beads and enriched, as in library selections. The enriched population was assessed by flow cytometry. The epitope tag stain differentiated pHLA-E-expressing yeast from competitor yeast.

### Library design and selection
Randomized peptide libraries were generated using polymerase chain reaction (PCR) on the pMHC construct with primers encoding NNK degenerate codons (N = any base; K = G or T), to encode 9 or 10 randomized amino acids. Primer sequences are provided in Supplemental Data 2.

Randomized pMHC PCR product was mixed with linearized pYAL vector backbone at a 5:1 mass ratio and electroporated into electro-competent RJY100 yeast[66]. The final 9mer and 10mer libraries contained approximately $1.5 \times 10^8$ and $7 \times 10^7$ yeast transformants, respectively.

As described above, yeast were subject to selections for CD94/NKG2x binding. Selections were performed using streptavidin-coated magnetic beads (Miltenyi Biotec; Bergisch Gladbach, Germany) coupled to biotinylated CD94/NKG2x. Yeast were cultured, induced, and selected in three iterative rounds of selection. In a fourth round of selections, yeast were incubated with tetrameric AlexaFluor647-conjugated streptavidin (produced in-house) coupled to biotinylated CD94/NKG2x, and selected with ∝-AlexaFluor647 magnetic beads (Miltenyi Biotec). Each round was preceded by negative selection clearance round with uncoated streptavidin beads (rounds 1–3) or streptavidin and ∝-AlexaFluor647 beads (round 4). In round four, prior to addition of beads and selection, a sampling of CD94/NKG2x

tetramer-stained yeast were assessed via flow cytometry on an Accuri C6 flow cytometer.

To prevent contamination with clonal yeast, the library template DNA encoded a stop codon in the peptide-encoding region. Additionally, the C-terminal HA tag was swapped for a FLAG tag to readily assess library expression and contamination.

### Library sequencing and analysis
Following selections, plasmid DNA from ~$10^7$ yeast were extracted using a Zymoprep II Yeast Miniprep Kit (Zymo Research; Irvine, CA) from each round of selection and the unselected library. Amplicons for deep sequencing were generated by PCR using the purified plasmids from yeast miniprep. Two rounds of PCR were completed to add homology for sequencing primers, i5 and i7 paired-end handles, and sequencing barcodes that are unique to each round of selection and selection reagent, to enable pooling of DNA. Primer sequences are provided in Supplemental Data 2. Amplicons were sequenced on an Illumina MiSeq (Illumina; San Diego, CA) with $2 \times 150$ nt paired-end reads at the MIT BioMicroCenter.

Paired-end reads were assembled and filtered for length and correct flanking sequences using PandaSeq[67]. To correct for PCR or sequencing errors, and given the immense sampling space of peptides, sequences were clustered with more frequent sequences within Hamming Distance = 1 in DNA space using CDHit[68,69]. Peptide sequences were translated from DNA and stop codon-containing sequences removed using an in-house script.

Supplemental Data 1 contains processed peptide data. In this file, sequences not containing stop codons are listed with their read counts, labeled for round of selection and selection reagent. In the column header, the letter indicates the selection reagent, and number indicates the selection round (e.g. "A1" is post-round 1 of selection with NKG2A). "R0" is the unselected round 0 library, and cross-selected libraries are labeled with their original selection reagent followed by the current selection reagent (e.g. "CA4" is post-round 4 of selection using NKG2A on the library that was previously selected with NKG2C). To filter for cross-contamination during library preparation, peptides included in Supplemental Data 1 were removed from the unselected round 0 library dataset if they contain greater than five reads.

### Heat map and sequence logo visualization of library-enriched peptides
Sequence logo visualizations were generated using a custom script (adapted from[70]) which is similar to the WebLogo[71] sequence logo generator webtool, without limiting the number of input peptides.

Heatmaps were generated with a custom script, on all enriched peptides from a given round of selection. Proportions were calculated by weighting each peptide by its read count, and the relative proportions of each amino acid at each position are shown. Maps were binned as shown to capture the frequency of reads that encode a peptide with a given amino acid at a given position.

Statistical analysis of enriched peptides was performed using Two Sample Logo[72], as done previously for pMHC-II binding[29]. To use Two Sample Logo, a subsample of ten thousand peptides, weighted by read count were used, with positive samples from Round 3 libraries and negative samples from the unselected library. Motifs and analysis were performed using the Two Sample Logo defaults, excepting significance was defined using two-sided binomial test with Bonferroni correction. The cutoff for significance was $p < 0.05$.

### Clustering peptides to identify subdominant motifs
We identified the subdominant cluster of WN peptides using various clustering methods, including Gibbs Cluster 2.0[73] on peptides enriched in Round 3 of CD94/NKG2A selections (not weighted by read counts), excluding stop codon-containing peptides, using the default method

"MHC class I ligands of same length", modified to assess 1–15 clusters, with the trash cluster to remove outliers turned off.

## Prediction algorithm generation

CD94/NKG2A- or CD94/NKG2C-specific prediction models were trained on yeast display library data, with positive examples drawn from Round 3 of selections, with counts greater than or equal to 100, each appearing in the training data *int(count/100)* times. This filtering results in 376 and 399 unique positive examples, for NKG2A- and NKG2C-specific predictors, respectively. Negative examples were selected from the unselected library, with read count equal to 1 and excluding stop codon-containing peptides. The total number of negative peptides were selected such that there were equal numbers of positive and negative examples in the training set (total number, not unique examples), resulting in 12,135 and 13,559 for NKG2A and NKG2C predictors, respectively. Positive examples were assigned a target value of 1 and negative examples were assigned a target value of 0.

These data were used to train NNAlign 2.0 with MHC Class I defaults, excepting Maximum Length for Deletions, Maximum Length for Insertions, and Length of PFR for Composition Encoding were set to zero; Encode PFR Length was set to −1 (for no encoding); and both Binned Peptide Length Encoding and Length of Peptides Generated from FASTA Entries were set to 9.

## Prediction on proteome-derived peptides

With a 9mer window, step size 1, 11,019,710 unique human proteome 9mers were generated from a reference proteome (Uniprot UP000005640), excluding peptides containing selenocysteine. The same was done for human CMV (reference Uniprot proteome UP000000938), generating 61,303 peptides. UL120$_{71-79, AD169}$ and UL120$_{71-79, BE/33/2010}$ were derived from sequences in Uniprot UP000008991 and UP000100992, respectively.

A one-sided Mann–Whitney U test was performed on HLA-E-binding VL9 signal peptides from the human proteome compared to other 9mer human proteome peptides (11,019,710 total 9mer peptides). $U$-values were 66,115,459 (NKG2A) and 66,118,060 (NKG2C) and were calculated alongside $p$-values using scipy version 1.4.1 in Python version 3.7.3.

## NetMHC predictions

NetMHCpan4.1 predictions were performed with a local version of the algorithm on 9mer peptides for HLA-E*01:03.

## Recombinant protein production

Recombinant soluble HLA-E single chain trimers, CD94/NKG2A, and CD94/NKG2C for SPR were produced using a baculovirus expression system with High Five (Hi5) insect cells (ThermoFisher, catalog number B85502). Individual constructs were cloned into pAcGP67a vectors. For each, 2 μg of plasmid DNA was transfected into SF9 insect cells (ThermoFisher, catalog number 11496015) using BestBac 2.0 baculovirus DNA (Expression Systems; Davis, CA) and Cellfectin II reagent (ThermoFisher). Viruses were propagated to high titer and transduced into Hi5 cells, grown at 27 °C for 48–72 h, and purified from pre-conditioned cell media supernatant using Ni-NTA resin and size exclusion chromatography with a S200 column on an AKTAPure FPLC (GE Healthcare; Chicago, IL).

Single chain timers were formatted with the C-terminus of the peptide linked to the N-terminus of human β2M via a flexible linker. In turn a flexible Gly-Ser linker connects the C-terminus of β2M to the N-terminus of the extracellular α1, α2, and α3 domains of HLA-E*01:03 heavy chain, containing a Y84A mutation to accommodate the linked peptide. An AviTag biotinylation tag and poly-histidine tag were connected to the C-terminus of HLA-E heavy chain. Protein was biotinylated overnight before FPLC-based purification.

CD94/NKG2x proteins were expressed as single-chain fusion proteins. The C-terminus of human NKG2A or NKG2C ectodomain was connected via a GGSGGS linker to human CD94 ectodomain. The C-terminus of CD94 is connected to an AviTag biotinylation tag and poly-histidine tag. Protein for SPR was used fresh and dialyzed overnight into HBS-EP+ buffer (10 mM HEPES pH 7.4, 150 mM NaCl, 3 mM EDTA, and 0.05% v/v Surfactant P20) (Cytiva; Malborough, MA) or buffer exchanged into HBS-EP+ buffer during FPLC-based purification before use in SPR. CD94/NKG2x for selections was expressed from the Expi293 expression system (ThermoFisher, catalog number A14527) as per the manufacturer's recommendations, using a modified pHLSec-Avitag3-His6 vector[74], and purified by IMAC (HisTrap Excel column) and SEC (S200 Increase GL 10/30) utilizing an Akta Pure (Cytiva). Biotinylation was confirmed by streptavidin gel shift assay[75].

## Surface plasmon resonance

Steady-state surface plasmon resonance experiments were performed with a Biacore T200 instrument. CD94/NKG2x in HBS-EP+ buffer was injected as analyte in a concentration range of 0.1 μM to 102.4 μM and flow rate of 10 μL/min at 25 °C. Data was fit with Prism 9.4 (GraphPad Software Inc; San Diego CA) to a "one site, specific binding" model. Sensorgrams are included in Supplemental Fig. 13.

For investigating WN peptides, biotinylated pMHC was immobilized at approximately 400 Response Units (RU) on a Series S SA sensor chip (Cytiva). WN peptide (WNRILPNAY) HLA-E single chain trimer (SCT), VMAPRTLFL peptide HLA-E SCT, and reference HLA-DR401 (HLA-DRA1*01:01, HLA-DRB1*04:01, linked to CLIP peptide)[29] were immobilized. CD94/NKG2A was injected followed by CD94/NKG2C. Approximate $K_D$ values calculated by Biacore software are as follows: WN SCT+CD94/NKG2A: 41.5 μM; WN SCT+CD94/NKG2C: 17.4 μM; VMAPRTLFL SCT+CD94/NKG2A: 8.2 μM; and VMAPRTLFL SCT+CD94/NKG2C: 21.4 μM.

For investigating human-derived peptides, Series S CM5 chips (Cytiva) were coupled to neutravidin and utilized. Biotinylated pMHC was immobilized at approximately 400 RU. On one chip, VMAPRTLFL HLA-E SCT, BFAR$_{263-271}$ HLA-E SCT, CREB3L1$_{419-427}$ HLA-E SCT, and reference HLA-DR401 were immobilized. CD94/NKG2A was injected followed by CD94/NKG2C. On a second chip, PISD$_{55-63}$ HLA-E SCT, AKAP6$_{388-396}$ HLA-E SCT, GTF3C5$_{293-301}$ HLA-E SCT, and reference HLA-DR401 were immobilized. CD94/NKG2C was injected followed by CD94/NKG2A. $K_D$ values calculated by Biacore software are reported in Fig. 5. $R_{max, fit}$ values were calculated by Biacore software. $R_{max, theoretical}$ values were calculated using the analyte/ligand mass ratio (0.66) multiplied by the amount of ligand coupled to the chip.

## Prometheus differential scanning fluorimetry experiments

Differential scanning fluorimetry (DSF) experiments were performed with a Prometheus NanoTemper NT.48 (Munich, Germany) to measure protein stability, using intrinsic tryptophan fluorescence.

For these experiments, refolded empty HLA-E was generated. Human β2M plus a poly-histidine tag and HLA-E*01:03 extracellular α1, α2, and α3 domains plus an AviTag biotinylation tag were separately codon optimized for *E. coli* and cloned into the bacterial expression vector pET28a. Inclusion bodies for HLA-E and β2M were made in BL21 *E. coli*. Inclusion bodies were purified, homogenized, and dissolved, including denaturation in 8 M urea solution. Before refolding, each IB was mixed with an equal volume of Gdn-Cl solution (6 M Guanidine HCl, 500 mM Tris, 2 mM EDTA, 100 mM NaCl) and incubated overnight at 37 °C. β2M was injected dropwise with a 27-gauge needle into refolding buffer (100 mM pH 8.3 Tris, 400 mM L-arginine hydrochloride, 2 mM EDTA, 0.5 mM oxidized glutathione, 5 mM reduced glutathione, 0.2 mM PMSF) and incubated, stirring gently, at 4 °C for 1 h, after which an equal mass of HLA-E heavy chain was similarly injected. This solution incubated at 4 °C overnight, followed by two

days of dialysis in 10 mM Tris + 50 mM NaCl. Protein was concentrated in 10 kDa molecular weight cutoff centrifugal concentrators and purified by size exclusion chromatography on an AKTAPURE FPLC (GE Healthcare, Chicago IL), using an S200 column followed by a S75 column. Protein was concentrated with size filters as initial attempts to concentrate using the poly-histidine tag and Ni-NTA resin resulted in the protein precipitating out of solution.

Individual DSF reactions were set up with 9 μg of MHC and 100×, 50×, or 10× peptide to MHC ratio by molarity (peptides from GenScript). The final reaction volume was 20 μL and reaction mixtures were incubated at room temperature for 20–30 min.

Prometheus excitation power was set such that raw fluorescence counts were between 8000–15000 for each sample. Samples were heated at 1 °C per minute, from 20–95 °C. DSF measurements were taken in duplicate with high-sensitivity and standard capillaries. Melting temperatures were similar for a given peptide across conditions, and representative data from 50x ratio in high sensitivity capillary are presented. A secondary inflection point around 60 °C in most melts is consistent with melting of β2M[76].

### Cell lines
K562/HLA-E[44] (obtained from E. Weiss) and RMA-S/HLA-E[48] (obtained from J. Coligan) were maintained in complete medium (RPMI-1640 containing glutamine and supplemented with 10% v/v FBS, 20 μM β-mercaptoethanol and 100 U/mL penicillin-streptomycin; all Thermo Fisher) in the presence of 400 μg/mL hygromycin B and 1 mg/mL G418 (both InvivoGen), respectively.

### HLA-E surface stabilization assay
RMA-S/HLA-E cells were incubated with serial dilutions of peptides (3–300 μM, Genscript) in OptiMEM (ThermoFisher) for 16 h at 37 °C. Cells were washed with PBS, stained for HLA-E (Clone 3D12, Biolegend, 1:200 dilution) for 15 min at room temperature and analyzed by flow cytometry (analysis in FlowJo version 10). Peptide UBAC2$_{275-279}$ was ordered as a 5mer peptide (WNRLF) given the similarity with other WN peptides and hypothesis of a truncated peptide binding in the HLA-E groove. Due to lower peptide solubility, the maximal concentration tested for BFAR$_{263-271}$ was 150 μM, for EMC1$_{725-733}$ was 75 μM, for OR5D14$_{88-96}$ was 30 μM, and for UL102$_{292-300}$ was 30 μM. Measurements from replicate experiments are plotted in Fig. 3 and Supplemental Fig. 8.

### NK cell activation assays
All analyses of human data were carried out in compliance with the relevant ethical regulations. Healthy blood donors gave informed consent at DRK Blutspendedienst Nord-Ost, Dresden, Germany, and buffy coats were obtained as approved by Charité ethics committee (EA4/059/17). PBMCs were isolated from human CMV-seropositive buffy coats by density centrifugation over Ficoll Paque Plus (GE Healthcare) and screened for the presence of NKG2C$^+$ NK cell expansions by flow cytometry to have sufficient frequencies of single-positive cells for parallel assessment of activation and inhibition via HLA-E[47]. Total NK cells from donors with NKG2C$^+$ expansions (>10% NKG2A$^-$/NKG2C$^+$ NK cells) were enriched with CD56 microbeads (Miltenyi Biotec) and cryopreserved in FBS with 10% v/v DMSO. After thawing, NK cells were washed with complete medium, stained with antibodies and Fixable Viability Dye eFluor 780 (Thermo Fisher) for 15 min at 4 °C in PBS, sorted as viable CD56$^+$ CD3$^-$ cells, and rested in complete media overnight. RMA-S/HLA-E or K562/HLA-E were pulsed with the indicated peptides at a concentration of 300 μM as described above (150 μM for BFAR$_{263-271}$ due to lower peptide solubility). NK cells were co-cultured with target cells at an effector-to-target ratio of 2:1 in complete media in the presence of 300 μM peptide (150 μM for BFAR$_{263-271}$) and anti-CD107a. Brefeldin A and Monensin (BD Biosciences) were added after a 1 h culture period. After an additional 5 h,

the stimulation was stopped by centrifugation at 4 °C. Cells underwent surface staining, followed by fixation and intracellular staining using Inside Stain Kit (Miltenyi Biotec). Samples were analyzed by flow cytometry on an LSR Fortessa (BD Biosciences). Inhibition of NKG2A$^+$ NK cells and activation of NKG2C$^+$ NK cells was assessed by gating on the respective single-positive populations (Supplemental Fig. 14). Antibodies are listed in Supplemental Table 1: anti-human CD56 PE/Dazzle 594 (1:200 dilution), CD3 APC-eFluor 780 (1:50 dilution), anti-human CD159a (NKG2A) Biotin REAfinity (1:50 dilution), anti-human CD159c (NKG2C) PE REAfinity (1:100 dilution), anti-human CD107a (LAMP-1) Alexa Fluor 488 (1:400 dilution), anti-human IFN-γ PE-Vio 770 REAfinity (1:50 dilution), anti-human TNF-α Brilliant Violet 605 (1:50 dilution), anti-human CCL3 (MIP-1α) REAfinity (1:25 dilution).

### Data availability
All next-generation sequencing data generated in this study are available on the Sequence Read Archive (SRA) with accession code PRJNA859187. Processed peptide data are provided in Supplemental Data 1. Source data are provided with this paper.

Referenced proteomes were accessed via Uniprot with accession codes UP000005640, UP000000938, UP000008991, and UP000100992.

### Code availability
Scripts for data processing and analysis and NNAlign model files are publicly available at https://github.com/birnbaumlab/Huisman-et-al-2022-HLA-E.

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

## Acknowledgements
We would like to thank Lucy Walters for helpful discussions about HLA-E, Max Quastel for generously sharing HLA-E Prometheus protocols, and the MIT BioMicro Center for library sequencing. This work was supported in part by the Koch Institute Support (core) Grant P30-CA14051 from the National Cancer Institute. This work was supported by National Institute of Health (U19-AI110495), the Melanoma Research Alliance Foundation, and the Packard Foundation to M.E.B; a National Science Foundation Graduate Research Fellowship to B.D.H; Deutsche Forschungsgemeinschaft (DFG) grants SPP 1937 (RO3565/4-2) and SFB TRR241 B02 to C.R.; Leibniz-Science Campus Chronic Inflammation and Leibniz-Kooperative Exzellenz K259/2019 to C.R.; Berlin Health Innovations (BHI) Validation Fund to C.R. and T.R.; the Bill and Melinda Gates Foundation grant OPP1133649 to G.M.G; and the Medical Research Council grant MR/M019837/1 to G.M.G.

## Author contributions
Project conception: B.D.H., N.G., M.E.B.; Conducted experiments: B.D.H., N.G., T.R., L.G., N.K.S.; Performed data analyses: B.D.H., N.G., T.R.; Supervised the work: A.J.Mc.M., G.G., C.R., M.E.B.; Wrote the manuscript: B.D.H., M.E.B.; All authors contributed to the editing of the manuscript.

## Competing interests
B.D.H., T.R., C.R., and M.E.B. are listed as inventors on a patent application (PCT/US23/68073) covering peptides identified in this study. M.E.B. is an equity holder in 3T Biosciences, and is a co-founder, equity holder, and consultant of Kelonia Therapeutics and Abata Therapeutics. The remaining authors declare no competing interests.
