## [Peer Review File · Nature Communications]

High-throughput characterization of HLA-E-presented CD94/NKG2x ligands reveals peptides which modulate NK cell activationREVIEWER COMMENTS

Reviewer #1 (Remarks to the Author):

This is a very well written study about investigating the peptide repertoire that is presented by HLA-E and recognized by the receptors CD94/NKG2A and CD94/NKG2C. By utilizing a yeast-displayed peptide library, the authors identify ~500 high-confidence peptides, which they use to train a prediction algorithm, which was then applied to the human and huCMV proteomes. The thus predicted peptides were tested for HLA-E binding and their functional effect on NK cells having either of the two CD94/NKG2x receptors. The authors identified peptides modulating NK activity, in particular some peptides that were exclusively activating.

As the authors point out, the results are of significance because the HLA-E peptide repertoire is not yet well characterized, and NK activity-modulating peptides could have therapeutic applications.

Overall, the study is sound, the methodology is adequate and the conclusions are fair. Some important points are however only mentioned in the methods or in supplemental files, and should be explicitly mentioned in the main text.

Below are a few suggestions to strengthen the manuscript:

Perhaps the authors can find a better description than "HLA-E and CD94/NKG2x peptide repertoire" to describe the fact that they mean peptides that are presented by HLA-E and recognized by CD94/NKG2x?

As the authors even point out that in their 10mer library, the tenth residue acted as part of the linker, it is really important that the limits of the SCT system are discussed in the discussion. The respective sentence in the methods (p 18, line 363ff) should be removed and also included into the discussion.

The authors should clearly state how many of their yeast selected peptides were included in the training of the algorithm (e.g. on page 7, in the methods, and maybe also in the discussion. Along these lines, they should also include a comparison of their approach with the HLA-E prediction that is provided by NetMHCpan into the discussion. (As their source data are much larger, and not only bind to HLA-E but also to two receptors, this would even be beneficial for them).

Page 11, first line (209): It is very strange that peptides containing "disfavored residues that are unique to non-binders" show up among the top-10 (human) or top-3 (CMV). This seems to be a flaw in the algorithm. Please discuss this phenomenon.

In the discussion, the authors should also clearly point out another limitation of their study, namely that it is purely in vitro and in silico work (yeast display, predictions, assays with peptides pulsed onto cells). Thus, the HLA-E presented peptides could be different in vivo. This should at least be discussed, but the authors may also consider teaming up with an immunopeptidomics group, to assess if any of their proteome-derived peptides can be directly identified on HLA-E molecules. Such data would massively strengthen their conclusions and the impact of their work.

Minor:

Page 5, text describing Suppl. Fig. 2B: The results for the staining with the NKG2C tetramer should also be described. In particular as it is quite strange that the staining after 3 rounds of selection with this molecule is about at the level of the negative control. This should be mentioned and discussed.

Page 9, last line: Please indicate the rank that the divergent peptide achieved in your algorithm.

Description of Figure 4: As the color code from Figure 3 is carried through to Figure 4, the behavior of the different groups defined in Fig. 3 should be mentioned in the description of Fig. 4. (E.g. the fact that all the peptides that led to the best HLA-E surface stabilization also resulted in NK activity inhibition, but that there was no effect of HLA-E surface stabilization group on NK activation.)

Description of Fig. 5B: Please explain the difference in normalizing to fitted R_{max} or theoretical R_{max} , so that a reader who has never worked with surface plasmon resonance can interpret the figure.

Methodological:

It is not clear why the most recent version of NetMHCpan, which is NetMHCpan4.1, was not used. The authors should repeat their HLA-E prediction with the newest version, and also use the reference for the newest version.

Supplemental Figure 4: Please include the sequence logos under the respective heatmaps and also rearrange the figure so that heatmaps and logos from the same round are always next to each other. It would also be helpful for the reader if you could indicate in the respective text (page 6, line 139) in which of the panels it can be seen that the motif is less diverse for NKG2A.

Figure 3 onwards: Why was the known CMV peptide not included as a control in these experiments?

It is not clear to a reader who is not routinely dealing with NK cells how the NKG2A⁻/NKG2C⁺, NKG2A⁺/NKG2C⁻ and NKG2A⁻/NKG2C⁻ populations can be derived from healthy donors, especially as it is described in the methods that only cells from donors with "NKG2C expansions" were used. Could you please include a sentence in the respective method section on how these receptors normally occur on NK cells?

Very minor:

Results, line 93: Please describe that you mean the HA epitope tag also in the text, so that the reader can understand Figure 1C more easily.

Results, line 97: The tetramer-positive cells in Fig 1C can hardly be described as a population. Consider rephrasing to "positively stained cells" or something similar.

Page 11, line 225 onwards: The peptide previously termed "VL9" is now referred to as "LFL". Please either change this or explain why a different term is used.

Please replace "activity-skewed" by "activity-modulating" (or "activity-enhancing" where appropriate)
Page 18, line 350: Should read "MHC-E presented" (not "presenting")

Reviewer #2 (Remarks to the Author):

In the manuscript by Birnbaum and co-workers an innovative approach is used to identify HLA-E-binding peptides that also bind CD94/NKG2A or CD94/NKG2C. The authors screen a yeast library displaying fully randomized 9- or 10-mer peptides conjugated in cis via flexible linkers to B2-microglobulin and the HLA-E*01:03[Y84A] heavy chain ectodomain connected to yeast Aga2 cell surface protein. Yeast display libraries were selected with high-avidity biotinylated recombinant CD94/NKG2x receptors bound to magnetic streptavidin-coated beads and additionally with CD94/NKG2x tetramers having low avidity. As a technical flaw, in contrast to the NKG2A tetramer the NKG2C tetramer appears to have such a low affinity (Suppl. Fig. 1) that no enrichment can be demonstrated for CD94/NKG2C-selected yeasts using the latter staining tool (Suppl. Fig. 2B, lower panel).

After 3 rounds of selection clear peptide motifs emerged in a deep sequencing approach that showed high similarities between peptides displayed by single-chain trimer selected with either CD94/NKG2A or CD94/NKG2C with a great degree conservation of peptide positions 5-9 in tethered 9-mer peptides reflecting the known interface of pHLA-E and CD94/NKG2x. The same peptide motif is confirmed with a tethered 10-mer/HLA-E library, suggesting that 10-mers are bound in the same register with P2 and

P9 as main anchors and a non-conserved P10 residue extends out the groove. The authors note a strong preference for proline at P3, while the known canonical anchor P2 (Leu/Met) is less pronounced in the 9-mer libraries, albeit being more obvious in the 10-mer library. The authors note the novelty of their finding as to the strong preference for proline at P3 in the HLA-E peptide binding motif and discuss this result in the light of discrepant previous literature (p. 16). The well-known publicly available MHC peptide prediction tool NetMHC (<https://services.healthtech.dtu.dk/service.php?NetMHCpan-4.1>; Motif viewer), however, exactly shows this P3 proline preference for naturally presented peptides that have been eluted from HLA-E*01:03 (and HLA-E*01:01), which is also more dominant than the preference for leucine at the canonical P2 anchor position. Hence, the herein reported P3 proline preference of HLA-E should be discussed as confirming a known HLA-E peptide binding motif. Remarkably, peptide motifs derived from yeast peptide display libraries at the same time perfectly mirrored known features of the HLA-E[VMAPRTLFL]-CD94/NKG2A interface as revealed by crystal structures (Petrie et al., 2008; Kaiser et al., 2008), with a dominant interaction of CD94/NKG2A with Arg at P5 and Phe at P8. Taken together, the peptide motifs shown in Fig. 2 and Suppl. Fig. 6 provide convincing evidence that the yeast display approach used by Huisman et al. is highly effective. In my opinion, this represents a remarkable technical progress.

Next a self-designed, previously published neural network algorithm is trained with the results from NKG2A- and NKG2C-selected yeast peptide display in order to predict HLA-E AND NKG2x-binding peptides within a reference human proteome as well as within proteomes of cytomegalovirus strains. Judged by the properties of the 10 best-ranked binders derived from the NKG2A- and NKG2C-based algorithms, respectively, this bioinformatics approach appears to be successful with regard to CD94/NKG2x binding as all displayed peptides contain Arg at P5 and most contain Phe, Trp or Leu at P8 (Table 2), while with regard to HLA-E binding there appears to be a wide range of peptide affinities ranging from excellent binders (25.49 nM, RMPPRSLL, INTS1, #8 NKG2C) to non-binders with extremely poor affinities (45671.89 nM, WNRLFPLR, UBAC2, #6 NKG2A), as predicted by this reviewer with the help of NetMHC 4.1b. Hence, the applied NNAlign architecture appears to be of very limited predictive value concerning stable HLA-E binders. The order of stabilization of surface-expressed HLA-E in TAP-deficient RMA-S cells (Fig. 3) by algorithm-retrieved peptides correlates (with the exception of the TACR3 peptide containing an unusual cysteine at P8 that could lead to dimerization) with high accuracy with the order of NetMHC-predicted affinities (%Rank_EL etc.) and has therefore, in my view, a rather trivial explanation, again not speaking in favor of the application of NNAlign for the prediction of HLA-E peptide ligands. The authors could have achieved the goal of retrieving a panel of some 15-20 bona fide high-affinity HLA-E AND NKG2x binders from the human proteome by simply feeding the degenerate motif X-[LM]-[PA]-X-R-[ST]-[LVMI]-[FWLI]-[LFM] into the publicly available ScanProsite tool (www.prosite.expasy.org).

As a corollary of the inaccurate prediction of high-affinity HLA-E binders by the NNAlign algorithm, a number of peptides were ranked highly (Table 2) that possess a suboptimal P2 anchor residue associated with decreased predicted affinities (NetMHC 4.1b; all reported peptides assessed by this reviewer) and stabilization of HLA-E (Fig. 3). The three activation-skewed peptides pointed out by the authors, PISD(53-63, TAPARTMFL, 2293 nM), BFAR(263-271, VNPGRSLFL, 4423 nM) and GTF3C5(293-301, TGPWRSLWI, 8084 nM), fall in this category. While I cannot exclude that activation-skewed HLA-E-binding peptides may exist, that are detectable in functional assays with NKG2C/A+ NK cells, this conclusion seems to be premature as peptides PISD and GTF3C5 completely failed to activate NKG2C+ human NK cells when presented by RMA-S/HLA-E cells (the RMA-S/HLA-E inhibition assay with NKG2A+ NK cells unfortunately is missing in Fig. 4). It seems possible that co-stimulatory receptors expressed by K562, but not by murine RMA-S cells, lowered the threshold for activation. Furthermore, an HLA-E cell surface staining of K562/HLA-E and RMA-S/HLA-E cells would have been helpful to judge their respective capacities to present HLA-E ligands.

In accordance with poor HLA-E affinities, even single-chain trimers of BFAR/HLA-E[Y84A] and GTF3C5/HLA-E[Y84A] seem to be unstable as suggested by the SPR affinity measurements shown in Fig. 5. Possibly, the respective tethered peptides dissociate from the HLA-E heavy chain and do not rebind during the SPR assay. This could have been tested by using respective disulfide-trapped SCT for comparison. In my opinion, the poor HLA-E stabilizing capacity of putative activity-skewed peptides does not advocate the therapeutic use of such peptides (lines 324/325). Furthermore, the problem of selective expansion of donor-derived NKG2C+ adaptive NK cells has recently been solved

sufficiently by using disulfide-trapped HLA-E single chain trimers loaded with the "VL9" peptide and expressed by appropriate feeder cells (Murad et al., IJMS, 2022).

Two of the UL120 CMV peptides containing a Lys at P5 (page 10) and that do not stabilize HLA-E (Fig. 3) behave unusually – one acting as a putative NKG2A agonist/NKG2C antagonist and one being silent in terms of NKG2x interaction. Hence, Lys at P5 may imply NK-suppressive functions in response to particular CMV strains. However, before speculating about functional consequences of UL120 variations for NK cells responses (p. 17/18) it needs to be demonstrated that the intermediate-affinity HLA-E binder UL120[HCMV Merlin, VLPHRTQFL] and the weak binders UL120[AD169, SAPLKTRFL] and UL120[BE/33/2010, SVPLKTRFL] are indeed proteasomally processed and sufficiently loaded onto HLA-E in vivo. The concentration of 30 μ M used for pulsing of HLA-E expressing APC (Fig. 4) tends to be artificially high and will lead to occupancies by peptide exchange reaction that are far beyond what is expectably achieved through natural processing and ER peptide loading in HCMV infection, especially in the presence of high-affinity competitors such as derived from HLA-I ER signal sequences or TAP-independent UL40 leader peptides. The "WN" peptide part of this report appears to rather be a bioinformatics artifact than useful for the advancement of NK cell therapy.

Reviewer #3 (Remarks to the Author):

Huisman et al define an improved HLA-E and NK receptor peptide repertoire and exploit it for the identification of sequences that maintain CD94/NKG2C activation without inhibiting via CD94/NKG2A. The novel insights in the HLA-E repertoire and the three identified activity-skewed ligands advance the field and can be expected to serve as reference for future research into HLA-E and NK cell regulation.

The authors validated the outcome by stabilization in vitro and in cells. They also investigated how selective activation is achieved and found that the determined binding affinities are not predictive. Experiments are conducted, described and discussed very well. The methodology is sound and exceeds the current standards of the field.

Major:

1. Despite the availability of differentially acting sequences and the availability of complementary read-outs it remains unclear how skewed signalling can be rationalized, screened or predicted. Clarifying this point will be valuable for the reader. Including to what extent the novel insights into the repertoire specifically enabled the design of the presented skewed binders and summarize how this could be achieved systematically. Further clarify how their observations on the Response Units (RU) in SPR together with the in vitro TMs indicate that the formed peptide-HLA-E complexes would be less stable. (lines 274-277)

Minor:

1. The title states unbiased characterization. The reader will benefit from an explanation in what way their analysis is less biased and how their design of the experiments and the data can rule out residual biases. Alternatively this wording could be removed from the title.

2. Since the heatmaps are given naturally without derivations its not clear whether color differences actually indicate statistical significant differences. Accordingly, more details on how exactly the heatmaps were generated will be helpful.

3. First, multiple rounds of display in yeast followed by selection achieve the recapitulation of known binding features for CD94/NKx binding. The 9mer display was largely recapitulated with 10mers. Here a "loose" 10th position is to be expected validating the initial result and further confirming that its indeed only a 9mer that is required for binding. Yet, the 10th position suggests a bias towards "F, H, S, A residues". The reader will benefit from a clarification of the relevance of this position and the sequence requirement for this position.

4. Related to (3.) - Only multiple rounds of selection achieved to reveal a number of unprecedented sequence requirements summarized in sequence logos in figure 2. Yet, a number of the tested sequences do not fully follow the sequence proposed in this logo. Vice versa, some sequences largely

following the obtained sequence preference turned out to be low-affinity binders or false positives such as UL120. Again the reader will benefit from clarification to what extent the logo in fig2 can be expected to recapitulate the entire HLA-E repertoire. To this end testing of negative sequences that display features of known or identified positives but miss identified key AAs could help. Alternatively the authors could elaborate on the limitation of the repertoire represented by the logo.

5. By probing NK cell activation the authors identified sequences on par with VL9. In addition, the same assay also identified a number of peptides that achieve NK cell activation without inhibition which should be valuable expanding NK cells for therapeutic use. These "activity-skewed" binders were further analysed via SPR. Here its not entirely clear which of the two fits was used for the shown Kds. In addition, the determined K_{ons} and K_{offs} could be added and it could be discussed how kinetic trends correlate or do not correlate with activities.

Reviewer #4 (Remarks to the Author):

The work describes a strategy based on yeast display to find peptides that can bind to HLA-E and be recognized by CD94/NKG2x. A motif is then extracted and used to screen CMV proteome. Some of the predicted ligands activate NK cells.

HLA-I motifs have been extensively studied for HLA-A/-B/-C, but less is known about HLA-E peptides, and especially those recognized by CD94/NKG2x. The work provides a nice contribution to fill this gap, and the technology could be used for other purposes.

Major comments:

- The motif that has been found corresponds to a convolution between the HLA-E motif and the one required for recognition by CD94/NKG2x. This should be better emphasized, with the pros (predicted peptides will not only bind HLA-E, but also be recognized by CD94/NKG2x) and the cons (there may be other role for HLA-E ligands, and the very strong signal coming from the recognition by CD94/NKG2x will prevent correctly predicting these peptides, even if they are well displayed on HLA-E). In particular, the current predictor is likely of limited relevance for HLA-E ligands possibly recognized by T cells. This also likely explains the fact that the 10-mers tend to show a motif corresponding the 9-mer + X. A very compelling story would be to generate the motif for HLA-E itself, irrespective of the NK receptor. The authors have the technology to do it (e.g., Huisman et al 2022), and this would add a lot to the manuscript and could be very useful for predicting HLA-E ligands potentially recognized by other receptors.

- Following my previous point, I would recommend analyzing more thoroughly what in the motif reflects binding to HLA-E and what reflects CD94/NKG2x. For instance, the preference for Pro at P3 may reflect HLA-E (as suggested for instance by the motif viewer of NetMHCpan), but this is not really consistent with other data, like the peptides listed in IEDB.

- A potential strong bias comes from the fact that peptides are covalently linked at the C-terminus in the yeast display. This is especially relevant in the case of peptides recognized by NK receptors, since they mostly bind the MHC and peptide close to the C-terminus of the peptide. This issue should be discussed and addressed

- Not clear why some sequences in supp have high counts at R0, and show the motif observed after selection. Could there be some bias in the initial library towards peptides that show the motif? Or contaminations in the pooled sequencing. This is important to sort out and make sure the data are clean.

Minor comment:

- Authors should be careful when speaking about "HLA-E-binding peptide repertoire", or "HLA-E and NK receptor peptide". As mentioned earlier, the work does not identify the HLA-E-binding peptide repertoire, but only a small subset of peptides that are also recognized by CD94 proteins. When

starting to read the manuscript I was quite confused. Also, the "and" in "HLA-E and CD94/NKG2x peptide repertoire" can be confusing – people could imagine two different motifs, one for HLA-E and one for CD94/NKGx. I would strongly advice to use an unambiguous formulation, like "Peptides displayed on HLA-E and targeted/recognized by CD94/NKG2x".

REVIEWER COMMENTS

We thank the reviewers for their careful review and thoughtful feedback. In this revision, we have addressed reviewer critiques, and we resubmit an expanded and strengthened manuscript. Included below are detailed responses to each point in blue. With this resubmission, we include a manuscript file with text edits denoted as tracked changes. A summary of these revisions includes:

- We have further characterized and contextualized our tetramer staining and tetramer selection data. This includes an expanded Supplemental Figure 1, containing new experimental data from orthogonal bead-based enrichment assays.
- We performed further computational analyses, including generating additional sequence logos and performing statistical analysis of enrichment, provided in Supplemental Figures 4 and 5.
- We have expanded our discussion to contextualize our work with respect to immunopeptidomics, recent work from Murad and coworkers, and NetMHC, as well as extensibility of our approach for future identification and design of NK cell activity-modulating peptides.
- We clarify and discuss the role and use of NNAlign with an expanded discussion section.
- We have edited language throughout to improve clarity.

Reviewer #1 (Remarks to the Author):

This is a very well written study about investigating the peptide repertoire that is presented by HLA-E and recognized by the receptors CD94/NKG2A and CD94/NKG2C. By utilizing a yeast-displayed peptide library, the authors identify ~500 high-confidence peptides, which they use to train a prediction algorithm, which was then applied to the human and huCMV proteomes. The thus predicted peptides were tested for HLA-E binding and their functional effect on NK cells having either of the two CD94/NKG2x receptors. The authors identified peptides modulating NK activity, in particular some peptides that were exclusively activating.

As the authors point out, the results are of significance because the HLA-E peptide repertoire is not yet well characterized, and NK activity-modulating peptides could have therapeutic applications.

Overall, the study is sound, the methodology is adequate and the conclusions are fair. Some important points are however only mentioned in the methods or in supplemental files, and should be explicitly mentioned in the main text.

Below are a few suggestions to strengthen the manuscript:

Perhaps the authors can find a better description than “HLA-E and CD94/NKG2x peptide repertoire” to describe the fact that they mean peptides that are presented by HLA-E and recognized by CD94/NKG2x?

We have altered our phrasing throughout the text (with alternate phrasing such as “HLA-E-presented CD94/NKG2x ligands”) to provide clarity.

As the authors even point out that in their 10mer library, the tenth residue acted as part of the linker, it is really important that the limits of the SCT system are discussed in the discussion. The respective sentence in the methods (p 18, line 363ff) should be removed and also included into the discussion.

Thank you for this suggestion. We have moved this sentence from the methods section to the discussion, to highlight features and limits of the single chain trimer system.

The authors should clearly state how many of their yeast selected peptides were included in the training of the algorithm (e.g. on page 7, in the methods, and maybe also in the discussion. Along these lines, they should also include a comparison of their approach with the HLA-E prediction that is provided by NetMHCpan into the discussion. (As their source data are much larger, and not only bind to HLA-E but also to two receptors, this would even be beneficial for them).

In the results and methods sections, we have added the number of peptides included in training the algorithm, resulting from the previously-described criteria for their selection.

NetMHCpan4.1 predicts the novel human and CMV peptides which we identify and validate here to be binders, consistent with our data and with many of our identified peptides having canonical P2 and P9 anchors (e.g. P2 Met and P9 Leu are highly represented). Given our yeast display data and predictors are looking at peptides that can bind a composite of MHC and NK receptor, we would expect that a good NetMHC score is not sufficient to predict CD94/NKG2x binding. That is, CD94/NKG2x binders and non-binders alike likely comprise top predicted binders. We also include additional discussion comparing our motifs with the motif represented by the NetMHC motif viewer.

Page 11, first line (209): It is very strange that peptides containing “disfavored residues that are unique to non-binders” show up among the top-10 (human) or top-3 (CMV). This seems to be a flaw in the algorithm. Please discuss this phenomenon.

In seeking to understand why these peptides (named in former line 209) did not stabilize HLA-E appreciably in the RMA-S assay (**Figure 3, Supplemental Figure 10**), we looked for differences between the stabilizing and non-stabilizing peptides. The first peptide (UBAC2₂₇₅₋₂₇₉ [WNRLF]) contains a P1 Trp, P2 Asn motif, which we explore in **Supplemental Figure 6, Supplemental Figure 10**, and elsewhere, concluding that the “WN” motif-containing peptides appear to be an artifact of the single chain trimer system. The remaining three peptides (EMC1₇₂₅₋₇₃₃ [VMGDRSVLY], OR5D14₈₈₋₉₆ [VMADKSIFY], and UL102₂₉₂₋₃₀₀ [TGAARSFFF]) contain P7 Phe or P9 Tyr, which are absent from stabilizing peptides. P7 Phe and P9 Tyr appear in yeast display-enriched peptides, though, notably, among the most enriched peptides containing P7 Phe or P9 Tyr are peptides also containing the WN motif (e.g. WNRILPNAY and WNHILDFGR). Additionally, all three of EMC1₇₂₅₋₇₃₃, OR5D14₈₈₋₉₆, and UL102₂₉₂₋₃₀₀ lack P3 Pro, a residue that is present in all of the stabilizing peptides, except for FBXO41₆₇₀₋₆₇₈, which contains some of the top residues at each of the other positions. With these differences in stabilizing and non-

stabilizing peptides in mind, the single chain trimer-specific enrichment of “WN” peptides and the underweighting of important positional residue covariation (e.g., the requirement for P3 Pro unless all other residues are optimal) has resulted in the prediction of these non-binding peptides. We have expanded the text to highlight these points.

In the discussion, the authors should also clearly point out another limitation of their study, namely that it is purely in vitro and in silico work (yeast display, predictions, assays with peptides pulsed onto cells). Thus, the HLA-E presented peptides could be different in vivo. This should at least be discussed, but the authors may also consider teaming up with an immunopeptidomics group, to assess if any of their proteome-derived peptides can be directly identified on HLA-E molecules. Such data would massively strengthen their conclusions and the impact of their work.

Thank you. We have highlighted this limitation in our discussion. Natural processing is certainly of interest, although immunopeptidomics have been limited in characterization of the HLA-E peptidome because signal from highly abundant VL9 peptides dominates (Lee et al, Journal of Immunology 1998), with non-VL9 peptides comprising only a minor fraction of detected species (McMurtrey et al, PLoS ONE 2017), alongside potential noise such as highly-charged and long peptides (McMurtrey et al, PLoS ONE 2017; Kraemer et al, Stem Cells International 2015). In future work, as the detection limits in mass spectrometry continue to advance, comparing the HLA-E eluted peptidome with our identified HLA-E and CD94/NKG2x binders would be of interest.

Minor:

Page 5, text describing Suppl. Fig. 2B: The results for the staining with the NKG2C tetramer should also be described. In particular as it is quite strange that the staining after 3 rounds of selection with this molecule is about at the level of the negative control. This should be mentioned and discussed.

We have expanded the text to discuss the staining and will highlight here as well. Round 4 library staining with CD94/NKG2x tetramer is consistent with the levels of tetramer-positive staining on yeast expressing a canonical VL9 peptide presented by HLA-E (**Supplemental Figure 1A**). Differences between CD94/NKG2A and CD94/NKG2C staining also aligns with their affinity differences; for example, **Figure 5c** highlights the ~3-fold weaker affinity of CD94/NKG2C for VMAPRTLFL/HLA-E compared to CD94/NKG2A. Additionally, the minimal CD94/NKG2C tetramer staining is consistent with previous reports of tetramer-negative staining with low affinity receptors (Sabatino et al, JEM 2011), and matches our experience with other pMHC/immune receptor interactions where peptide sequences with measurable yet weak interactions may bind via beads but not tetramers (e.g., Birnbaum et al, Cell 2014). This highlights the need for high-avidity selection reagents, which motivated our use of MACS beads for selection Rounds 1-3. The low avidity of tetramers and low CD94/NKG2C affinity for pHLA-E combine to generate low tetramer signal.

Because of the low tetramer staining, similarities between motifs of post-Round 3 peptides and CD94/NKG2C-selected post-Round 4 peptides, and similar hierarchy of peptides between

CD94/NKG2C tetramer and Round 3 selections (**Supplemental Figure 3**), the CD94/NKG2C tetramer selection likely did not further enrich the library. As such, we did not utilize the tetramer data for algorithm training, opting instead to train our predictor on the bead-based selection data.

We have conducted additional column enrichment experiments using CD94/NKG2x conjugated to high avidity magnetic beads (shown in the expanded **Supplemental Figure 1**). With yeast expressing VMAPRTLFL/HLA-E or post-selection libraries, we stained these yeast with an antibody against a construct-expressed epitope tag and doped them into irrelevant yeast that do not express HLA-E. We then performed enrichments using CD94/NKG2A or CD94/NKG2C and see high-efficiency enrichment using high avidity magnetic beads.

Page 9, last line: Please indicate the rank that the divergent peptide achieved in your algorithm.

We have added the ranks of these peptides in the paper.

Description of Figure 4: As the color code from Figure 3 is carried through to Figure 4, the behavior of the different groups defined in Fig. 3 should be mentioned in the description of Fig. 4. (E.g. the fact that all the peptides that led to the best HLA-E surface stabilization also resulted in NK activity inhibition, but that there was no effect of HLA-E surface stabilization group on NK activation.)

Thank you for the suggestion: we have noted the origin of the color scheme in the **Figure 4** legend and related supplemental figures.

Description of Fig. 5B: Please explain the difference in normalizing to fitted R_{\max} or theoretical R_{\max} , so that a reader who has never worked with surface plasmon resonance can interpret the figure.

We have clarified this distinction in the text. R_{\max} is maximum change in responsive units in SPR, and a fitted R_{\max} is determined based on the saturation point or predicted saturation point of the measured curve. In contrast, the theoretical R_{\max} can be calculated based on the differences in mass between the immobilized protein and analyte protein which is flowed over, and is the theoretical maximum response units if all immobilized protein is bound. When the measured R_{\max} is less than theoretical, it may be due to less stability of the one of the interacting proteins, such that the effective concentration is lower than the measured concentration.

Methodological:

It is not clear why the most recent version of NetMHCpan, which is NetMHCpan4.1, was not used. The authors should repeat their HLA-E prediction with the newest version, and also use the reference for the newest version.

Thank you. We have updated the predictions (binding classifications are unchanged between these versions) and reference for the most recent version of NetMHCpan.

Supplemental Figure 4: Please include the sequence logos under the respective heatmaps and also rearrange the figure so that heatmaps and logos from the same round are always next to each other. It would also be helpful for the reader if you could indicate in the respective text (page 6, line 139) in which of the panels it can be seen that the motif is less diverse for NKG2A.

We have generated sequence logos corresponding to all of the heatmaps in **Supplemental Figure 4**. Of note, we also switched to generating our sequence logos using custom code, in place of the WebLogo web interface which limits the number of input sequences. Previously, in order to meet limitations of the WebLogo server, we scaled the number of peptides down by a factor of 100. Because the distribution of read counts in early rounds of selection is flatter (more peptides with few reads) and scaling is ineffective for capturing this spread, we utilized custom code for custom generation of sequence logos, utilizing all peptides in a round, weighted by their read counts. For consistency across figures, we have generated all of our sequence logos with this code; the previous sequence logos are largely identical to their replacements, and we have made this code available in our GitHub repository.

Additionally, we have added text to clarify which panels are less diverse for NKG2A, as suggested by the reviewer.

Figure 3 onwards: Why was the known CMV peptide not included as a control in these experiments?

We include three known CMV-derived peptides, VMAPRTLIL, VMAPRTLFL, and VMAPQSLLL (Hammer et al, Nature Immunology 2018; Heatley et al, J Biol Chem 2013) as controls in **Figure 4**, where VMAPRTLIL is the UL40-derived peptide predicted in **Table 3**. VMAPRTLFL was included in Figure 3; because VMAPRTLFL and VMAPRTLIL share MHC-facing residues, and their single amino acid difference is at the up-facing P8 residue, we expect the same HLA-E stabilization effects.

It is not clear to a reader who is not routinely dealing with NK cells how the NKG2A⁻/NKG2C⁺, NKG2A⁺/NKG2C⁻ and NKG2A⁻/NKG2C⁻ populations can be derived from healthy donors, especially as it is described in the methods that only cells from donors with “NKG2C expansions” were used. Could you please include a sentence in the respective method section on how these receptors normally occur on NK cells?

Thank you for highlighting the need for clarity on this point: the proportion of NKG2A⁺ and NKG2C⁺ NK cells vary widely across donors and disease states. For example, NKG2A is typically expressed in ~20-80% of NK cells (Mahaweni et al, 2018, Front. Immunol), and NKG2C⁺ NK cells expand upon CMV infection (Gumá et al., 2004, Blood), which is highly prevalent in healthy individuals (Cannon et al., 2010, Rev Med. Virol.). Because of this broad spread in NKG2A/NKG2C prevalence, donors are screened to ensure they have detectable NKG2C⁺ populations (>10 % NKG2A⁻/NKG2C⁺ NK cells) to ensure sufficient frequencies of single-positive cells for parallel assessment of activation and inhibition via HLA-E. Inhibition of NKG2A⁺ NK cells and activation of NKG2C⁺ NK cells was assessed by gating on the respective single-positive populations (gating strategy shown in **Supplemental Figure 13**). That is,

NKG2A⁺/NKG2C⁻, NKG2A⁻/NKG2C⁺, and NKG2A⁻/NKG2C⁻ populations were distinguished by flow cytometry. We have clarified these points in the text.

Very minor:

Results, line 93: Please describe that you mean the HA epitope tag also in the text, so that the reader can understand Figure 1C more easily.

We have clarified in the text that we are referring to the hemagglutinin (HA) epitope tag, and added clarification to the Figure 1 legend.

Results, line 97: The tetramer-positive cells in Fig 1C can hardly be described as a population. Consider rephrasing to “positively stained cells” or something similar.

We have made this change.

Page 11, line 225 onwards: The peptide previously termed “VL9” is now referred to as “LFL”. Please either change this or explain why a different term is used.

Thank you; we have updated the text for uniformity.

Please replace “activity-skewed” by “activity-modulating” (or “activity-enhancing” where appropriate)

We have made this change.

Page 18, line 350: Should read “MHC-E presented” (not “presenting”)

We have made this change.

Reviewer #2 (Remarks to the Author):

In the manuscript by Birnbaum and co-workers an innovative approach is used to identify HLA-E-binding peptides that also bind CD94/NKG2A or CD94/NKG2C. The authors screen a yeast library displaying fully randomized 9- or 10-mer peptides conjugated in cis via flexible linkers to B2-microglobulin and the HLA-E*01:03[Y84A] heavy chain ectodomain connected to yeast Aga2 cell surface protein. Yeast display libraries were selected with high-avidity biotinylated recombinant CD94/NKG2x receptors bound to magnetic streptavidin-coated beads and additionally with CD94/NKG2x tetramers having low avidity. As a technical flaw, in contrast to the NKG2A tetramer the NKG2C tetramer appears to have such a low affinity (Suppl. Fig. 1) that no enrichment can be demonstrated for CD94/NKG2C-selected yeasts using the latter staining tool (Suppl. Fig. 2B, lower panel).

As highlighted by the reviewer, the low affinity of CD94/NKG2C results in undetectable tetramer staining on our libraries which have undergone three prior rounds of bead-based selection. However, this is consistent with the low/undetectable binding to the canonical

VL9/HLA-E complex (**Supplemental Figure 1A**). Prior studies have also reported tetramer-negative staining for low affinity pMHC-receptor interactions, despite measurable binding via orthogonal assays (Sabatino et al, JEM 2011), and this is consistent with our experience with other pMHC/immune receptor interactions where peptide sequences with measurable yet weak interactions may bind via beads but not tetramers (e.g., Birnbaum et al, Cell 2014).

Differences between CD94/NKG2A and CD94/NKG2C staining also aligns with their affinity differences; for example, **Figure 5C** highlights the ~2.7-fold weaker affinity of CD94/NKG2C for VL9/HLA-E compared to CD94/NKG2A. The low CD94/NKG2C tetramer staining motivated our use of high avidity MACS beads for selections in rounds 1-3.

Because of the low tetramer staining, the CD94/NKG2C tetrameric selection in Round 4 likely did not further enrich the library beyond its Round 3 composition. As such, we did not utilize the tetramer data for algorithm training, opting instead to train our predictor on the bead-based selection data.

We have conducted additional column enrichment experiments using CD94/NKG2x conjugated to high avidity magnetic beads (shown in the expanded **Supplemental Figure 1**). With yeast expressing VL9/HLA-E or post-selection libraries, we stained these yeast with an antibody against a construct-expressed epitope tag and doped them into irrelevant yeast that do not express HLA-E. We then performed enrichments using CD94/NKG2A or CD94/NKG2C and see high-efficiency enrichment using high avidity magnetic beads.

After 3 rounds of selection clear peptide motifs emerged in a deep sequencing approach that showed high similarities between peptides displayed by single-chain trimer selected with either CD94/NKG2A or CD94/NKG2C with a great degree conservation of peptide positions 5–9 in tethered 9-mer peptides reflecting the known interface of pHLA-E and CD94/NKG2x. The same peptide motif is confirmed with a tethered 10-mer/HLA-E library, suggesting that 10-mers are bound in the same register with P2 and P9 as main anchors and a non-conserved P10 residue extends out the groove. The authors note a strong preference for proline at P3, while the known canonical anchor P2 (Leu/Met) is less pronounced in the 9-mer libraries, albeit being more obvious in the 10-mer library. The authors note the novelty of their finding as to the strong preference for proline at P3 in the HLA-E peptide binding motif and discuss this result in the light of discrepant previous literature (p. 16). The well-known publicly available MHC peptide prediction tool NetMHC (<https://services.healthtech.dtu.dk/service.php?NetMHCpan-4.1>; Motif viewer), however, exactly shows this P3 proline preference for naturally presented peptides that have been eluted from HLA-E*01:03 (and HLA-E*01:01), which is also more dominant than the preference for leucine at the canonical P2 anchor position. Hence, the herein reported P3 proline preference of HLA-E should be discussed as confirming a known HLA-E peptide binding motif. Remarkably, peptide motifs derived from yeast peptide display libraries at the same time perfectly mirrored known features of the HLA-E[VMAPRTLFL]–CD94/NKG2A interface as revealed by crystal structures (Petrie et al., 2008; Kaiser et al., 2008), with a dominant interaction of CD94/NKG2A with Arg at P5 and Phe at P8. Taken together, the peptide motifs shown in Fig. 2 and Suppl. Fig. 6 provide convincing evidence that the yeast display approach used by Huisman et al. is highly effective. In my opinion, this represents a remarkable technical progress.

Thank you. We were encouraged to see similarities between our yeast display-identified peptides and existing datasets on peptide-HLA-E binding. We have added a comparison to the NetMHCpan-4.1 motif in our Discussion section.

Next a self-designed, previously published neural network algorithm is trained with the results from NKG2A- and NKG2C-selected yeast peptide display in order to predict HLA-E AND NKG2x-binding peptides within a reference human proteome as well as within proteomes of cytomegalovirus strains. Judged by the properties of the 10 best-ranked binders derived from the NKG2A- and NKG2C-based algorithms, respectively, this bioinformatics approach appears to be successful with regard to CD94/NKG2x binding as all displayed peptides contain Arg at P5 and most contain Phe, Trp or Leu at P8 (Table 2), while with regard to HLA-E binding there appears to be a wide range of peptide affinities ranging from excellent binders (25.49 nM, RMPPRSPLL, INTS1, #8 NKG2C) to non-binders with extremely poor affinities (45671.89 nM, WNRLFPLR, UBAC2, #6 NKG2A), as predicted by this reviewer with the help of NetMHC 4.1b. Hence, the applied NNAlign architecture appears to be of very limited predictive value concerning stable HLA-E binders. The order of stabilization of surface-expressed HLA-E in TAP-deficient RMA-S cells (Fig. 3) by algorithm-retrieved peptides correlates (with the exception of the TACR3 peptide containing an unusual cysteine at P8 that could lead to dimerization) with high accuracy with the order of NetMHC-predicted affinities (%Rank_EL etc.) and has therefore, in my view, a rather trivial explanation, again not speaking in favor of the application of NNAlign for the prediction of HLA-E peptide ligands. The authors could have achieved the goal of retrieving a panel of some 15-20 bona fide high-affinity HLA-E AND NKG2x binders from the human proteome by simply feeding the degenerate motif X-[LM]-[PA]-X-R-[ST]-[LVMI]-[FWLI]-[LFM] into the publicly available ScanProsite tool (www.prosite.expasy.org).

It is important to note here that nearly all of our predicted proteome-derived peptides are also predicted by NetMHCpan4.1 to be HLA-E*01:03 binders, with the single exception of the mentioned UBAC2-derived peptide (WNRLFPLR). Of note, UBAC2₂₇₅₋₂₇₉ is an anomaly because it contains the N-terminal “WN” (P1 Trp, P2 Asn) motif that we explore within the paper (**Supplemental Figure 6** and **Supplemental Figure 10**). Peptides with a “WN” motif appear to be an artifact of the single chain trimer system, demonstrating measurable binding to CD94/NKG2x in the context of the HLA-E single chain trimer (**Supplemental Figure 6**) but not demonstrating binding to HLA-E in our stabilization assays with unlinked peptide (**Supplemental Figure 10**). As such, we see a strong concordance between our yeast display-trained model and NetMHCpan4.1 predictions for HLA-E binding, and characterize the properties of the peptide which is the exception (UBAC2₂₇₅₋₂₇₉). We have expanded our discussion to further highlight this caveat.

We also have clarified in our text that while our prediction algorithm enabled the selection of peptides described in the paper including peptides with NK cell activity-modulating behavior, it is not a unique solution. That is, other alternate model architectures may be trained on our underlying yeast display data to perform predictions. Indeed, we believe that one of the biggest impacts of our work is the underlying yeast display data that enables these predictions, such that the NNAlign framework could be exchanged for an alternate prediction model like the ScanProsite tool.

And, finally, we predict a composite of HLA-E binding and CD94/NKG2x binding, so that simply applying NetMHCpan4.1 to the human or CMV proteomes may select peptides for HLA-E stabilization (including most of those predicted by our models) but would not discern which of them can bind to CD94/NKG2x nor prioritize them among other predicted peptides.

As a corollary of the inaccurate prediction of high-affinity HLA-E binders by the NNAlign algorithm, a number of peptides were ranked highly (Table 2) that possess a suboptimal P2 anchor residue associated with decreased predicted affinities (NetMHC 4.1b; all reported peptides assessed by this reviewer) and stabilization of HLA-E (Fig. 3). The three activation-skewed peptides pointed out by the authors, PISD(53-63, TAPARTMFL, 2293 nM), BFAR(263-271, VNPGRSLFL, 4423 nM) and GTF3C5(293-301, TGPWRSLWI, 8084 nM), fall in this category. While I cannot exclude that activation-skewed HLA-E-binding peptides may exist, that are detectable in functional assays with NKG2C/A+ NK cells, this conclusion seems to be premature as peptides PISD and GTF3C5 completely failed to activate NKG2C+ human NK cells when presented by RMA-S/HLA-E cells (the RMA-S/HLA-E inhibition assay with NKG2A+ NK cells unfortunately is missing in Fig. 4). It seems possible that co-stimulatory receptors expressed by K562, but not by murine RMA-S cells, lowered the threshold for activation. Furthermore, an HLA-E cell surface staining of K562/HLA-E and RMA-S/HLA-E cells would have been helpful to judge their respective capacities to present HLA-E ligands.

An important caveat is that we did not perform selections to identify the highest affinity peptide-HLA-E binders, but rather peptides that bind both to HLA-E and to CD94/NKG2x. Because we are selecting on this composite of HLA-E and CD94/NKG2x binding, we expect variation in peptide-HLA-E binding affinity, as long as both HLA-E and NK receptor binding interactions can occur, which is consistent with the range of NetMHC predictions described by the reviewer (excepting the UBAC2 peptide discussed above).

We have edited the text for additional clarity about the RMA-S/HLA-E assay (**Figure 4B**). RMA-S cells do not activate NK cells in the absence of peptide-HLA-E. That is, surface activation markers are ~0% for the VMAPQSLLL negative control peptide in **Figure 4B** (the equivalent of the dotted line that we show in **Figure 4A** plots is near 0%, overlapping with the x-axis in **Figure 4B**). Though low, NK cell activation markers in the presence of PISD₅₃₋₆₃ (TAPARTMFL) or GTF3C5₂₉₃₋₃₀₁ (TGPWRSLWI) are distinctly above baseline, e.g. with an average percent CD107a⁺ of 7.8% (PISD) and 11.1% (GTF3C5), compared to 1.1% (VMAPQSLLL). Albeit low, these data and K562 assay data support that these peptides activate NKG2C⁺ NK cells.

The reviewer also mentioned the absence of RMA-S/HLA-E inhibition assay for NKG2A⁺ NK cells. Since the RMA-S cell lines don't activate NK cells in the absence of peptide-HLA-E (i.e. surface activation markers are close to 0% with the VMAPQSLLL negative control in the RMA-S assay), there's no equivalent of the NKG2C⁺ NK cells/RMA-S cell stimulation assay with NKG2A⁺ NK cells that can be conducted. In other words, because of the low baseline (starting with inactive NK cells), we cannot detect inhibition via NKG2A with RMA-S/HLA-E cells. Whereas, by contrast, K562 cells, which can activate NK cells in the absence of HLA-E

expression/NKG2C engagement (**Figure 4A**), enable us to look at both activation via NKG2C and inhibition via NKG2A.

HLA-E cell surface staining of the RMA-S/HLA-E cells is utilized to investigate HLA-E surface stabilization and shown in **Figure 3**.

In accordance with poor HLA-E affinities, even single-chain trimers of BFAR/HLA-E[Y84A] and GTF3C5/HLA-E[Y84A] seem to be unstable as suggested by the SPR affinity measurements shown in Fig. 5. Possibly, the respective tethered peptides dissociate from the HLA-E heavy chain and do not rebind during the SPR assay. This could have been tested by using respective disulfide-trapped SCT for comparison. In my opinion, the poor HLA-E stabilizing capacity of putative activity-skewed peptides does not advocate the therapeutic use of such peptides (lines 324/325). Furthermore, the problem of selective expansion of donor-derived NKG2C⁺ adaptive NK cells has recently been solved sufficiently by using disulfide-trapped HLA-E single chain trimers loaded with the “VL9” peptide and expressed by appropriate feeder cells (Murad et al., *IJMS*, 2022).

As the reviewer highlights, in our SPR measurements, the BFAR₂₆₃₋₂₇₁/HLA-E and GTF3C5₂₉₃₋₃₀₁/HLA-E single chain trimers appear unstable due to the lower measured response units compared to the theoretical R_{max}. This is consistent with our direct assay of HLA-E stabilization in a separate assay, presented in **Figure 3**, as well as data on BFAR₂₆₃₋₂₇₁ in **Supplemental Figure 9**. We have tempered our language around the potential therapeutic uses of these peptides, including highlighting potential future studies.

Thank you for highlighting the work from Murad et al, and we have added a citation to this paper. There are distinctions between our work and that from Murad et al which are important to note. Murad and coworkers showed the favored expansion of NKG2C⁺ NK cells using a prostate carcinoma cell line engineered to express IL-2, membrane-bound IL-15, and VL9/HLA-E single chain trimer. This presents an exciting opportunity expand NK cells from peripheral blood, including selective expansion of NKG2C⁺ cells. The effect of their feeder line to preferentially expand NKG2C⁺ cells is even more intriguing given the effects of VL9/HLA-E to signal through both NKG2A and NKG2C, with a predominant inhibitory effect; the combination of expressed cytokines and the cancer-derived feeder cells likely couple with VL9/HLA-E to contribute to this preferential expansion of NKG2C⁺ cells. Our work provides an orthogonal approach, identifying peptides which modulate NK cell activity, including a subset of peptides which preferentially activate NKG2C⁺ NK cells, when the peptide is presented untethered by two different antigen presenting cell lines, including the RMA-S cell line which is otherwise inert to NK cells. This demonstrates and identifies peptides which have preferential signaling through NKG2A or NKG2C. We present evidence that NKG2C⁺ cells can be selectively activated in a peptide-specific manner and provide data suggesting these effects are independent of their receptor affinity. Our work also presents avenues to explore for potential use of selectively activating peptides as a peptide vaccine therapy.

Two of the UL120 CMV peptides containing a Lys at P5 (page 10) and that do not stabilize HLA-E (Fig. 3) behave unusually – one acting as a putative NKG2A agonist/NKG2C antagonist and one being silent in terms of NKG2x interaction. Hence, Lys at P5 may imply NK-

suppressive functions in response to particular CMV strains. However, before speculating about functional consequences of UL120 variations for NK cells responses (p. 17/18) it needs to be demonstrated that the intermediate-affinity HLA-E binder UL120[HCMV Merlin, VLPHRTQFL] and the weak binders UL120[AD169, SAPLKTRFL] and UL120[BE/33/2010, SVPLKTRFL] are indeed proteasomally processed and sufficiently loaded onto HLA-E *in vivo*. The concentration of 30 μM used for pulsing of HLA-E expressing APC (Fig. 4) tends to be artificially high and will lead to occupancies by peptide exchange reaction that are far beyond what is expectably achieved through natural processing and ER peptide loading in HCMV infection, especially in the presence of high-affinity competitors such as derived from HLA-I ER signal sequences or TAP-independent UL40 leader peptides. The “WN” peptide part of this report appears to rather be a bioinformatics artifact than useful for the advancement of NK cell therapy.

UL120_{71-79, AD169} (SAPLKTRFL) peptide, which negatively affects activation of both NKG2A⁺/NKG2C⁻ and NKG2A⁻/NKG2C⁺ subsets, also negatively affected our internal NKG2A⁻/NKG2C⁻ control (**Supplemental Figure 11**), which suggests this peptide has nonspecific effects, so we exclude it from additional analysis. The closely related UL120_{71-79, BE/33/2010} peptide (SVPLKTRFL) showed minimal effects on all assessed subsets of NK cells, suggesting it does not engage with CD94/NKG2x in the context of HLA-E. This is potentially due to the P5 Lysine in the peptide. We have expanded our discussion to highlight peptide processing and have toned down our language related to potential functional consequences of variation in UL120 peptides.

The concentrations of peptides utilized here have been demonstrated in establishing this system for RMA-S/HLA-E, K562/HLA-E, and 721.221 cell lines (Borrego et al, JEM 1998; Lee et al, JI 1998; Michaëlsson et al, JEM; Heatley et al, J Biol Chem 2013). Additionally, titration of the canonical leader peptides using the same experimental system has revealed the range of 10¹-10² μM can best discriminate the differential ability of peptides to stabilize HLA-E and activate/inhibit NK cells (Hammer et al. Nat Immunol 2018, Figure 1e and Supplemental Figure 1h,k). In additional HLA-E *in vitro* experiments, the low affinity of HLA-E binding peptides, coupled with likely stimulation that would be present *in vivo* like co-receptor binding and cytokines, has established the use of saturating concentrations of peptides for studying HLA-E stabilization (e.g., Walters et al, Cell Reports 2022; Barber et al, Eur. J. Immunol. 2022).

The N-terminal “WN” peptide motif is present in our selection data, and this subdominant motif is highlighted in **Supplemental Figure 6**. Because of the appearance of these peptides in our selection data, they are likely an artifact of the single chain trimer system rather than a bioinformatics artifact. Since “WN” peptides are in the training data, peptides with this motif can be predicted by our algorithm (i.e. UBAC₂₂₇₅₋₂₇₉). Follow-up experiments suggest that the “WN” peptides do not stabilize HLA-E when added exogenously (**Supplemental Figure 10**), but can bind to HLA-E and CD94/NKG2x as a single chain trimer (**Supplemental Figure 6**), supporting that it is an artifact from the single chain trimer system. As such, we agree that their utility may be limited and we make no suggestions about their potential uses. We have edited the text to clarify this point.

Reviewer #3 (Remarks to the Author):

Huisman et al define an improved HLA-E and NK receptor peptide repertoire and exploit it for the identification of sequences that maintain CD94/NKG2C activation without inhibiting via CD94/NKG2A. The novel insights in the HLA-E repertoire and the three identified activity-skewed ligands advance the field and can be expected to serve as reference for future research into HLA-E and NK cell regulation.

The authors validated the outcome by stabilization *in vitro* and in cells. They also investigated how selective activation is achieved and found that the determined binding affinities are not predictive. Experiments are conducted, described and discussed very well. The methodology is sound and exceeds the current standards of the field.

Major:

1. Despite the availability of differentially acting sequences and the availability of complementary read-outs it remains unclear how skewed signalling can be rationalized, screened or predicted. Clarifying this point will be valuable for the reader. Including to what extent the novel insights into the repertoire specifically enabled the design of the presented skewed binders and summarize how this could be achieved systematically. Further clarify how their observations on the Response Units (RU) in SPR together with the *in vitro* TMs indicate that the formed peptide-HLA-E complexes would be less stable. (lines 274-277)

Though we did not initially set out to discover peptides which could selectively activate NKG2C⁺ cells, we were excited by the observation that several of our predicted peptides were able to achieve this effect. From the similarities between the NKG2A and NKG2C receptors and their peptide motifs, as well as the lower affinity of NKG2C, it initially appeared that these receptors may be too similar for selective NKG2C-mediated activation. However, after identifying selectively activating peptides, we conducted experiments to characterize their features which may enable future rational prediction. Several features of these peptides stand out. Firstly, their selective NKG2C activation is not mediated by a higher affinity for NKG2C (**Figure 5**); in fact, these peptides have a range of affinities for both receptors, and their relative affinities for NKG2A compared to NKG2C is similar to other non-skewing peptides. Secondly, these selectively-activating peptides exhibit a lower ability to stabilize HLA-E (**Figure 3, Figure 5, Supplemental Figure 9**). Their lower HLA-E stabilizing ability is consistent with and attributes of their sequences: BFAR₂₆₃₋₂₇₁, PISD₅₅₋₆₃, and GTF3C5₂₉₃₋₃₀₁ have less preferred P2 residues (P2 Asn, P2 Ala, P2 Gly, respectively), compared to the more canonically preferred P2 Met or P2 Leu. Simultaneously, their NK receptor-facing residues are among the most preferred (P5 Arg, P8 Phe or P8 Trp). Future work can use these features for systematic and rational design of activity-modulating peptides. We have expanded the text to highlight these points.

In the *in vitro* stabilization assays (**Figure 3**), BFAR₂₆₃₋₂₇₁ and GTF3C5₂₉₃₋₃₀₁ demonstrate less stabilization of HLA-E, based on lower detection of HLA-E surface expression in the presence of these peptides (both are part of the group of cyan-colored peptides). However, when these two peptides are assessed by SPR (e.g. summary values in **Figure 5C**), we see that the pHLA-E affinity for CD94/NKG2x is the highest (BFAR₂₆₃₋₂₇₁) and lowest (GTF3C5₂₉₃₋₃₀₁) of the assessed peptides. Importantly, though, we see that these peptides exhibit a distinct pattern in the maximum response units measured by SPR. The maximum response units (R_{\max}) is determined

by either: 1) fitting the measured binding curve ($R_{\max, \text{fit}}$), and this will be close to the maximum measured response if the binding curve saturates; or 2) calculating the theoretical maximum response units ($R_{\max, \text{theoretical}}$), which is the calculated theoretical maximal response units if all of the immobilized protein is active and able to be bound by analyte protein. The maximal measured response units are lower than the theoretical maximum ($R_{\max, \text{theoretical}}$) for BFAR₂₆₃₋₂₇₁ and GTF3C5₂₉₃₋₃₀₁, which is consistent with a lower effective concentration of bindable protein, suggesting they form less stable pMHC complexes.

Minor:

1. The title states unbiased characterization. The reader will benefit from an explanation in what way their analysis is less biased and how their design of the experiments and the data can rule out residual biases. Alternatively this wording could be removed from the title.

We initially used “unbiased” in the title to highlight starting with a randomized peptide library in our yeast display screen, rather than a select set of peptides, which may introduce biases. However, we agree with the reviewer’s point, so to avoid ambiguity, we have incorporated this feedback and altered the title.

2. Since the heatmaps are given naturally without derivations its not clear whether color differences actually indicate statistical significant differences. Accordingly, more details on how exactly the heatmaps were generated will be helpful.

The heatmaps in **Figure 2** show the proportions of each amino acid at each position in enriched peptides. The fractions of amino acids are discretized on the white-blue color scale for easier visualization, and raw fractions are included in the provided Source Data file. As such, the heatmaps do not intrinsically show statistical significance of the enriched residues. We performed analysis to determine the statistical significance of these enrichments using Two Sample Logo (Vacic et al, Bioinformatics 2006), and have added a new **Supplementary Figure 5**. The magnitude of these changes resulted in most enriched positional preferences being statistically significant.

3. First, multiple rounds of display in yeast followed by selection achieve the recapitulation of known binding features for CD94/NKx binding. The 9mer display was largely recapitulated with 10mers. Here a "loose" 10th position is to be expected validating the initial result and further confirming that its indeed only a 9mer that is required for binding. Yet, the 10th position suggests a bias towards "F, H, S, A residues". The reader will benefit from a clarification of the relevance of this position and the sequence requirement for this position.

These preferences are representative of the selection data, but are likely a property specific to the single chain trimer system rather than for HLA-E itself. Since the peptide is covalently linked to the MHC, and this architecture requires a Y84A mutation to stabilize the linker, the C-terminal P Ω (“P10”) residue may be interacting with the HLA-E helices where it passes through the open groove as part of the linker, and these amino acid preferences may indicate a preferential fit for small (Ser, Ala) and small aromatic (His, Phe) residues.

4. Related to (3.) - Only multiple rounds of selection achieved to reveal a number of

unprecedented sequence requirements summarized in sequence logos in figure 2. Yet, a number of the tested sequences do not fully follow the sequence proposed in this logo. Vice versa, some sequences largely following the obtained sequence preference turned out to be low-affinity binders or false positives such as UL120. Again the reader will benefit from clarification to what extend the logo in fig2 can be expected to recapitulates the entire HLA-E repertoire. To this end testing of negative sequences that display features of known or identified positives but miss identified key AAs could help. Alternatively the authors could elaborate on the limitation of the repertoire represented by the logo.

As the reviewer highlighted, there are some differences between predicted peptides and the characterized motifs. One such example is a set of predicted peptides which do not stabilize HLA-E: EMC1₇₂₅₋₇₃₃ [VMGDRSVLY], OR5D14₈₈₋₉₆ [VMADKSIFY], and UL102₂₉₂₋₃₀₀ [TGAARSFFF]. One of the most prominent distinctions between these peptides and binders is their absence of P3 Pro, which is present in all of the stabilizing peptides, except for FBXO4₁₆₇₀₋₆₇₈, which contains some of the most prevalent residues at each of the other positions. We hypothesize that the motifs capture HLA-E and CD94/NKG2x binding preferences, but the prediction algorithm may underweight some of the necessary amino acid covariations, such as the need for P3 Pro unless all of the other residues are optimized. Similarly, several peptides largely match the motifs but show weak stabilizing activity, such as BFAR₂₆₃₋₂₇₁, PISD₅₅₋₆₃, and GTF3C5₂₉₃₋₃₀₁ (also discussed above), although they have less preferred, although detected, P2 residues in our motif characterization. Several binders depart from the most abundant residues in the motif, although most of the binders adhere to key residue preferences, such as P5 Arg and hydrophobic C-terminal residues. Because we were selecting on a composite of HLA-E and CD94/NKG2x binding, we expect some variation in HLA-E binding strength, as long as both binding interactions can be fulfilled. In sum, we largely observe that the motifs in **Figure 2** are an accurate characterization of HLA-E and CD94/NKG2x preferences (with the exception of WN peptides, which appear to be a feature of the single chain trimer system, and discussed in the manuscript and in response to the other reviewers), consistent with the results of stabilization assays, but that covariate relationships between residues are not fully captured by the prediction algorithm, allowing for the prediction of several peptides with non-ideal residues or residue combinations. We highlight these points in our expanded Discussion section.

5. By probing NK cell activation the authors identified sequences on par with VL9. In addition, the same assay also identified a number of peptides that achieve NK cell activation without inhibition which should be valuable expanding NK cells for therapeutic use. These "activity-skewed" binders were further analysed via SPR. Here its not entirely clear which of the two fits was used for the shown Kds. In addition, the determined kons and Koffs could be added and it could be discussed how kinetic trends correlate or do not correlate with activities.

K_D values were calculated by the Biacore software using R_{max, fit}. We can see some coarse dissociation rate relationships in the sensorgrams in **Supplemental Figure 12**, but we set up the SPR chips for steady state measurements, rather than for kinetic measurements; for example, for kinetic measurements, we would ideally utilize higher flow rates (e.g. 50 μL/min rather than 10 μL/min) and lower surface density (<50 RU rather than 400 RU). As such, we cannot accurately generate fits, and thus we do not make any claims about kinetic parameters.

Reviewer #4 (Remarks to the Author):

The work describes a strategy based on yeast display to find peptides that can bind to HLA-E and be recognized by CD94/NKG2x. A motif is then extracted and used to screen CMV proteome. Some of the predicted ligands activate NK cells.

HLA-I motifs have been extensively studied for HLA-A/-B/-C, but less is known about HLA-E peptides, and especially those recognized by CD94/NKG2x. The work provides a nice contribution to fill this gap, and the technology could be used for other purposes.

Major comments:

- The motif that has been found corresponds to a convolution between the HLA-E motif and the one required for recognition by CD94/NKG2x. This should be better emphasized, with the pros (predicted peptides will not only bind HLA-E, but also be recognized by CD94/NKG2x) and the cons (there may be other role for HLA-E ligands, and the very strong signal coming from the recognition by CD94/NKG2x will prevent correctly predicting these peptides, even if they are well displayed on HLA-E). In particular, the current predictor is likely of limited relevance for HLA-E ligands possibly recognized by T cells. This also likely explains the fact that the 10-mers tend to show a motif corresponding the 9-mer + X. A very compelling story would be to generate the motif for HLA-E itself, irrespective of the NK receptor. The authors have the technology to do it (e.g., Huisman et al 2022), and this would add a lot to the manuscript and could be very useful for predicting HLA-E ligands potentially recognized by other receptors.

Generating the motif of peptides which bind to HLA-E alone is certainly of interest. As the reviewer pointed out, we have experience and technology to generate peptide-MHC motifs, however the technology mentioned (Huisman et al 2022) is limited to MHC class II. We have been limited in extending the platform to MHC-I due to differences in the openness/closedness of the binding groove and length of peptides. We have included this in our discussion, along with a discussion of the convolution of HLA-E and NK receptor binding preferences (expounded in the reviewer's next point).

- Following my previous point, I would recommend analyzing more thoroughly what in the motif reflects binding to HLA-E and what reflects CD94/NKG2x. For instance, the preference for Pro at P3 may reflect HLA-E (as suggested for instance by the motif viewer of NetMHCpan), but this is not really consistent with other data, like the peptides listed in IEDB.

We have expanded our text to discuss the composite nature of the peptide motif, and highlight which residues preferences likely drive HLA-E or CD94/NKG2x interactions.

- A potential strong bias comes from the fact that peptides are covalently linked at the C-terminus in the yeast display. This is especially relevant in the case of peptides recognized by NK receptors, since they mostly bind the MHC and peptide close to the C-terminus of the peptide. This issue should be discussed and addressed

From structures of CD94/NKG2A in complex with HLA-E, the contacts between these two molecules are more central to the HLA-E groove, with peptide contacts at P5 and P8 (PDB 3CDG, 3CII). In contrast, other NK receptors, chiefly KIRs, bind pHLA complexes at the C-terminus of the peptide (e.g., PDB 4N8V). We would be interested in extending the pHLA / NK receptor selection paradigm to KIRs but are limited, as the reviewer has highlighted, because the C-terminal linker and opened HLA groove would likely disrupt the KIR binding interface. We have added further discussion of peptide-HLA-E linkage and the limitations of applying this system to other NK receptors.

- Not clear why some sequences in supp have high counts at R0, and show the motif observed after selection. Could there be some bias in the initial library towards peptides that show the motif? Or contaminations in the pooled sequencing. This is important to sort out and make sure the data are clean.

Post-selection and unselected library rounds were processed, and DNA libraries generated, simultaneously, likely resulting in minor cross-contamination when preparing and pooling barcoded libraries for sequences. However, contaminating reads counts are low and represent a minor fraction of the data (**Supplemental Figure 3**). For training our prediction models, we selected negative examples from the unselected library, with read count of 1 (Methods section “Prediction algorithm generation”), in order to avoid any training on any contaminating reads. Additionally, the greatest utility of the yeast display data is in its aggregate form, where the composite motif and library-trained algorithm have more utility than individually enriched sequences. As a filter for cross-contamination in our provided Supplemental Data, we have excluded peptides from the unselected, Round 0 dataset in Supplemental Data 1 if they contain greater than five reads.

Minor comment:

- Authors should be careful when speaking about “HLA-E-binding peptide repertoire”, or “HLA-E and NK receptor peptide”. As mentioned earlier, the work does not identify the HLA-E-binding peptide repertoire, but only a small subset of peptides that are also recognized by CD94 proteins. When starting to read the manuscript I was quite confused. Also, the “and” in “HLA-E and CD94/NKG2x peptide repertoire” can be confusing – people could imagine two different motifs, one for HLA-E and one for CD94/NKGx. I would strongly advice to use an unambiguous formulation, like “Peptides displayed on HLA-E and targeted/recognized by CD94/NKG2x”.

Thank you for highlighting the ambiguity of this wording. We have updated our phrasing to clarify.

Reviewer #1 (Remarks to the Author):

The authors have satisfactorily addressed all my comments.

Reviewer #2 (Remarks to the Author):

The manuscript by Michael E. Birnbaum and co-workers has been reviewed by four independent reviewers that all pointed out an interesting technical advancement in the development of an peptide-HLA-E yeast display systems in order to elucidate features in peptides that on the one hand are presented by HLA-E molecules and on the other hand bind to either of the two structurally closely related inhibitory NK cell receptor NKG2A/CD94 and activating receptor NKG2C/CD94, respectively. All reviewers raised serious points of criticism chiefly related to the performance of a novel prediction algorithm developed after the sequencing of NKG2x/CD94-selected pHLA-E yeast display libraries and the interpretation of functional data obtained with particular HLA-E ligand predicted by the algorithm.

As a general observation, the authors did not deem it necessary to conduct and present new experiments in order to corroborate weak data contained in the previous version. Hence the manuscript only contains minor improvements with clarifying textual changes and additional peptide logos according to the extensive suggestions of the reviewers. Major issues were merely verbally addressed in the letter of rebuttal. The rebuttal is able to mitigate some points of concern, however, cannot convince in its entirety leaving this reviewer with the distinct impression that this not a compelling, mature and conclusive story of significant novelty and interest for a wider scientific readership that would deserve publication in a top-ranking journal.

More specifically with regard to the points raised by this reviewer (and majorly overlapping with those of others) - while the sequencing of selected yeast display libraries accurately produced a composite peptide motif with correctly placed main and side anchor residues for HLA-E binding as well as the previously described residues P5-Arg and P8-Phe/Trp needed for efficient NKG2x/CD94 interaction, the neural network algorithm trained on the yeast display data appears to be flawed, which is also quite freely admitted by the authors in the letter of rebuttal. Because stable peptide binding to HLA-E single-chain trimers is a prerequisite for NKG2x selection, it is very difficult to understand why the algorithm puts out such a large proportion (15 out of 19 peptides tested at 3 or 30 μ M concentration; Fig. 2 & Figs. S8/S10) of low-affinity binders and even several non-binders which are unable to stabilize HLA-E molecules displayed at the cell-surface of TAP-deficient RMA-S cells.

Nevertheless, the algorithm predicted three HLA-E ligands (BFAR, PISD, GTF3C5), all having unusual residues at the P2 position, i.e. Asn, Ala and Gly, respectively, that possess the interesting property of being non-inhibitory for NKG2A+ NK cells while maintaining some activating activity for NKG2C+ NK cells. According to the response to reviewer #3, this has been a rather fortuitous finding. Despite this contention being repeated at least 5 times in the manuscript, in my opinion only with goodwill these three peptides can comprehensively be called selective activators of NKG2C+ NK cells. The 3 peptides all showed a reduced activation level (reduced means in 4 graphs of Fig. 4B, right column), but more importantly, only the high-affinity ligand BFAR(263-271) was able to significantly activate NKG2C+ NK cells when using mouse RMA-S/HLA-E cells as peptide presenters. Thus, the only point of significant immunological relevance (rather than technical novelty) of the entire manuscript virtually hinges on the positive evaluation of the single peptide, VNPGRSLFL. As no long-term stimulations of NKG2C+ NK cells from multiple healthy donors have been used to validate the finding, I am doubtful whether this peptide will make it into a much-desired expansion protocol for NKG2C+ adaptive NK cells which would give the manuscript an optimistic outlook.

Two reviewers raised the concern that the staining with NKG2C/CD94 tetramers after four

rounds of selection with NKG2C-loaded magnetic beads/tetramers is not above background level. Since simple tetramers stainings with streptavidin-Alexa647 are not the end of the line in the terms of options to increase avidity (streptavidin-PE plus anti-PE antibodies, APC-U-Load-dextramers being only two examples) the excuses for this obvious experimental shortcoming expressed in the letter of rebuttal are hardly convincing. I had asked to show HLA-E expression levels of K562/HLA-E and RMA-S/HLA-E. Unfortunately I cannot find these FACS stainings in the revised manuscript.

Reviewer #4 (Remarks to the Author):

My Comments have been addressed.

REVIEWER COMMENTS

We thank the reviewers for reviewing our revised manuscript. In this second revision, we have further highlighted the limitations of our algorithm and have better qualified claims of immunological relevance. We have generated and included data from two additional experiments: FACS data showing HLA-E expression levels of K562/HLA-E and RMA-S/HLA-E cell lines (included as Supplemental Figure 11), and multimerized staining of HLA-E-expressing yeast to demonstrate CD94/NKG2C-mediated binding (added to Supplemental Figure 1). Below, we provide a point-by-point response to remaining reviewer comments.

Reviewer #1 (Remarks to the Author):

The authors have satisfactorily addressed all my comments.

Thank you for your comments and time to review our manuscript.

Reviewer #2 (Remarks to the Author):

The manuscript by Michael E. Birnbaum and co-workers has been reviewed by four independent reviewers that all pointed out an interesting technical advancement in the development of an peptide-HLA-E yeast display systems in order to elucidate features in peptides that on the one hand are presented by HLA-E molecules and on the other hand bind to either of the two structurally closely related inhibitory NK cell receptor NKG2A/CD94 and activating receptor NKG2C/CD94, respectively. All reviewers raised serious points of criticism chiefly related to the performance of a novel prediction algorithm developed after the sequencing of NKG2x/CD94-selected pHLA-E yeast display libraries and the interpretation of functional data obtained with particular HLA-E ligand predicted by the algorithm.

As a general observation, the authors did not deem it necessary to conduct and present new experiments in order to corroborate weak data contained in the previous version. Hence the manuscript only contains minor improvements with clarifying textual changes and additional peptide logos according to the extensive suggestions of the reviewers. Major issues were merely verbally addressed in the letter of rebuttal. The rebuttal is able to mitigate some points of concern, however, cannot convince in its entirety leaving this reviewer with the distinct impression that this not a compelling, mature and conclusive story of significant novelty and interest for a wider scientific readership that would deserve publication in a top-ranking journal.

While we believe that some of the previously suggested experiments about natural peptide presentation are beyond the scope of this manuscript, we have striven to conduct experiments to further validate our yeast display system, and to more strongly caveat the data we present and conclusions drawn from our work. To address reviewer inquiries, we have added experimental data in **Supplemental Figure 1B-D** in the first revision, and have now included new data in **Supplemental Figure 1E-G** and **Supplemental Figure 11** in this version of the manuscript. We have also further expanded our text changes to better qualify our results.

More specifically with regard to the points raised by this reviewer (and majorly overlapping with those of others) - while the sequencing of selected yeast display libraries accurately produced a

composite peptide motif with correctly placed main and side anchor residues for HLA-E binding as well as the previously described residues P5-Arg and P8-Phe/Trp needed for efficient NKG2x/CD94 interaction, the neural network algorithm trained on the yeast display data appears to be flawed, which is also quite freely admitted by the authors in the letter of rebuttal. Because stable peptide binding to HLA-E single-chain trimers is a prerequisite for NKG2x selection, it is very difficult to understand why the algorithm puts out such a large proportion (15 out of 19 peptides tested at 3 or 30 μ M concentration; Fig. 2 & Figs. S8/S10) of low-affinity binders and even several non-binders which are unable to stabilize HLA-E molecules displayed at the cell-surface of TAP-deficient RMA-S cells.

The yeast display selection criteria drive the features of our enriched and predicted peptides: our yeast display selections were performed to enrich for peptides which can both bind to HLA-E and CD94/NKG2x. That is, we do not explicitly enrich for high affinity peptide-HLA binders, and our selections permit peptides which bind to HLA-E at a range of affinities, as long as they can be presented by HLA-E for binding by CD94/NKG2A or CD94/NKG2C. In orthogonal experimental systems, we observe that peptides with low HLA-E affinity (**Figure 3**) can show functional effects on NK cells (**Figure 4**), binding to CD94/NKG2x when presented by HLA-E (**Figure 5**), and HLA-E stabilization in a separate assay (**Supplemental Figure 9**). These results support that many of the weak HLA-E binders predicted by our algorithm are bona fide binders to HLA-E and CD94/NKG2x and represent true-positive predictions.

We would also like to clarify the distribution of peptides among these categories (**Figure 3**, **Supplemental Figure 8**, **Supplemental Figure 10**) as the description above does not fully capture the results of our stabilization assay:

- Four peptides show HLA-E stabilization matching VMAPRTLFL peptide (INTS₁₂₆₀₋₂₆₈, HLA-A₃₋₁₁, ECEL₁₂₆₉₋₂₇₇, TACR₃₂₂₆₋₂₃₄)
- Seven peptides show strong binding that is weaker than VMAPRTLFL (CREB3L₁₄₁₉₋₄₂₇, AKAP₆₃₈₈₋₃₉₆, UL120₇₂₋₈₀, Merlin, MTREX₄₉₀₋₄₉₈, FBXO4₁₆₇₀₋₆₇₈, SLC52A3₃₅₄₋₃₆₂, PISD₅₅₋₆₃)
- Two predicted peptides show weak but detectable stabilization (BFAR₂₆₃₋₂₇₁, GTF3C5₂₉₃₋₃₀₁). We assessed a further two CMV-derived peptides (UL120₇₁₋₇₉, AD₁₆₉, UL120₇₁₋₇₉, BE/33/2010) that were not in our top model predictions (**Table 2**, **Table 3**), which we included because of their similarity to a model-predicted peptide (UL120₇₂₋₈₀, Merlin).
- Four peptides showed no detectable binding (UBAC2₂₇₅₋₂₇₉, EMC1₇₂₅₋₇₃₃, OR5D14₈₈₈₋₉₆, UL102₂₉₂₋₃₀₀), which we explore with orthogonal experiments (e.g. 'WN' peptides in **Supplemental Figure 10**), and we discuss features of these peptides in the text (e.g. 'WN' motif, absent P3 Pro residue, or P7 Phe and P9 Tyr).

Nevertheless, the algorithm predicted three HLA-E ligands (BFAR, PISD, GTF3C5), all having unusual residues at the P2 position, i.e. Asn, Ala and Gly, respectively, that possess the interesting property of being non-inhibitory for NKG2A+ NK cells while maintaining some activating activity for NKG2C+ NK cells. According to the response to reviewer #3, this has been a rather fortuitous finding. Despite this contention being repeated at least 5 times in the manuscript, in my opinion only with goodwill these three peptides can comprehensively be called selective activators of NKG2C+ NK cells. The 3 peptides all showed a reduced activation level (reduced means in 4 graphs of Fig. 4B, right column), but more importantly, only the high-affinity ligand BFAR(263-271) was able to significantly activate NKG2C+ NK cells when using

mouse RMA-S/HLA-E cells as peptide presenters. Thus, the only point of significant immunological relevance (rather than technical novelty) of the entire manuscript virtually hinges on the positive evaluation of the single peptide, VNPGRSLFL. As no long-term stimulations of NKG2C⁺ NK cells from multiple healthy donors have been used to validate the finding, I am doubtful whether this peptide will make it into a much-desired expansion protocol for NKG2C⁺ adaptive NK cells which would give the manuscript an optimistic outlook.

We have toned down our claims of immunological relevance (at lines 193, 403, 412, and 439 in the revised manuscript). Please note that, although BFAR₂₆₃₋₂₇₁ shows the highest activation signature in NKG2C⁺ NK cells among the three described peptides in RMA-S/HLA-E activation assays as highlighted by the reviewer, all three of the indicated peptides (PISD₅₅₋₆₃, BFAR₂₆₃₋₂₇₁, and GTF3C5₂₉₃₋₃₀₁) have distributions of activation and degranulation markers which are significantly different compared to the control peptide VMAPQSLLL in RMA-S/HLA-E activation assays, excepting GTF3C5₂₉₃₋₃₀₁ by TNF expression, by the two-tailed t-test with a *p*-value cutoff of 0.05. With this in mind, we believe these peptides are accurately characterized as selectively activating. We also would like to emphasize that that selective signaling through NKG2C at any level above baseline, without signaling through NKG2A is noteworthy given 1) NKG2A and NKG2C have similar peptide binding preferences (**Figure 2**), and 2) NKG2C is lower affinity than NKG2A (**Figure 5**, Kaiser et al PNAS 2008).

We believe that, while interesting, the mechanistic and functional effects of long-term stimulation of NKG2C⁺ NK cells is beyond the scope of this work. We present data on, and restrict our claims to, the discovery of NKG2A/C-CD94 peptide binding motifs in the context of HLA-E, and the identification of peptides that show a skew in initial NK cell activation and degranulation, shown with four markers in **Figure 4**. Our description of selectively activating peptides underscores the existence of peptides which can selectively induce signaling via the lower-affinity NKG2C, provides a set of peptides whose features give us insights into characteristics which enable this skewed signaling, and investigates the mechanism for their effect (**Figure 5**). In our revised text, we have striven to qualify our interpretation of these peptides and their future outlook.

Two reviewers raised the concern that the staining with NKG2C/CD94 tetramers after four rounds of selection with NKG2C-loaded magnetic beads/tetramers is not above background level. Since simple tetramers stainings with streptavidin-Alexa647 are not the end of the line in the terms of options to increase avidity (streptavidin-PE plus anti-PE antibodies, APC-U-Load-dextramers being only two examples) the excuses for this obvious experimental shortcoming expressed in the letter of rebuttal are hardly convincing.

We have performed additional staining assays, included as **Supplemental Figure 1E-G**, using tetramers, dextramers, and tetramers further crosslinked by anti-streptavidin antibodies. Staining of post-Round 4 selected yeast shows a high degree of staining with all reagents, including tetramers, suggesting Round 4 tetramer selection was successful. The motif that arises from CD94/NKG2C enrichment experiments is consistent with both the data from CD94/NKG2A-based selections and the MHC- and NK receptor-binding motif. HLA-E/VMAPRTLFL showed similar degrees of staining among all three staining formulations. While we note that the binding of VMAPRTLFL remains below the limit of detection for NKG2C-based staining reagents, the

selective (if modest) staining of this pMHC with NKG2A-based reagents demonstrates that the yeast display construct is functional, while the library-based enrichment (including robust tetramer staining after four rounds) demonstrates that the CD94/NKG2C staining reagent is functional. The degree of staining is consistent with the previously observed tetramer data in both yeast display experiments and TCR-pMHC interactions for low affinity interactions, and supports our experimental decision to use highly avid CD94/NKG2x beads for selections. These results are consistent with some of our previous studies that show that streptavidin-coated beads are the most sensitive means of conducting these experiments, and can identify orthogonally-validated pMHC binders with a low false positive rate (as demonstrated in Birnbaum et al Cell 2014, where peptide motifs for binding to the 2B4 TCR including a P5 Ser and P8 Ile/Leu were selected via beads but did not enrich via tetramers, yet were demonstrated to bind and activate via a crystal structure of the TCR/pMHC complex, SPR, and functional assays). **Supplemental Figure 1B-D** from our prior revision experiment demonstrates binding of HLA-E to highly avid CD94/NKG2x-coated beads.

I had asked to show HLA-E expression levels of K562/HLA-E and RMA-S/HLA-E. Unfortunately I cannot find these FACS stainings in the revised manuscript.

We apologize for this omission. The expression levels of HLA-E on RMA-S/HLA-E cell lines comprises **Figure 3**. We have added a side-by-side comparison of K562/HLA-E and RMA-S/HLA-E levels as **Supplemental Figure 11**.

Reviewer #4 (Remarks to the Author):

My Comments have been addressed.

Thank you for your comments and time to review our manuscript.